# Oxytocin modulates local topography of human functional connectome in healthy men at rest

Daniel Martins[1], Ottavia Dipasquale [1] & Yannis Paloyelis [1✉]

Oxytocin has recently received remarkable attention for its role as a modulator of human behaviour. Here, we aimed to expand our knowledge of the neural circuits engaged by oxytocin by investigating the effects of intranasal and intravenous oxytocin on the functional connectome at rest in 16 healthy men. Oxytocin modulates the functional connectome within discrete neural systems, but does not affect the global capacity for information transfer. These local effects encompass key hubs of the oxytocin system (e.g. amygdala) but also regions overlooked in previous hypothesis-driven research (i.e. the visual circuits, temporal lobe and cerebellum). Increases in levels of oxytocin in systemic circulation induce broad effects on the functional connectome, yet we provide indirect evidence supporting the involvement of nose-to-brain pathways in at least some of the observed changes after intranasal oxytocin. Together, our results suggest that oxytocin effects on human behaviour entail modulation of multiple levels of brain processing distributed across different systems.

[1] Department of Neuroimaging, Institute of Psychiatry, Psychology and Neuroscience, King's College London, De Crespigny Park, London SE5 8AF, UK.
✉email: yannis.paloyelis@kcl.ac.uk

Oxytocin has received remarkable attention over the last decades for its role as a modulator of several physiological and cognitive processes in both human and non-human species[1,2]. Targeting the central oxytocin system has been proposed as a potential adjunctive treatment to improve outcome in several conditions, such as autism spectrum disorder[3], schizophrenia[4], eating disorders[5,6] or pain[7]. Small scale studies aiming to harness the oxytocin system through administration of exogenous oxytocin (either using nasal sprays or intravenous infusion) have shown beneficial effects in some disorders, although with mixed results[8]. Despite these recent advances, our knowledge about the neural mechanisms through which oxytocin affects human behaviour is still insufficient.

Most studies examining the effects of oxytocin on the human brain have used task-evoked functional magnetic resonance imaging (fMRI)[9]. Although these types of studies are important for examining specific cognitive or affective processes within a task-dependent context, the results are, by definition, constrained by the nature of the employed tasks. A paradigm-independent approach, such as resting state fMRI, offers the opportunity to uncover basic pharmacological mechanisms independently of the specific psychological processes being studied. Additionally, previous studies have shown that interindividual differences in the functional hard-wiring of the human brain at rest predict interindividual differences in task-evoked brain activity across a wide range of social, cognitive and affective processes[10]. Therefore, investigating the effects of oxytocin on the functional brain substrates engaged at rest may provide insights that help us understand its pharmacodynamics and its potential as an adjunctive treatment for several neuropsychiatric disorders.

Using arterial spin labelling fMRI, our previous work has uncovered the spatiotemporal pattern of changes in brain perfusion at rest, following the administration of oxytocin using various routes[11,12]. Changes in perfusion provide a quantitative, non-invasive pharmacodynamic marker of the local effects of acute doses of psychoactive drugs[13,14] with high-spatial resolution and excellent temporal reproducibility[15]. However, this approach cannot fully illuminate the effects of drugs on complex biological systems such as the brain, which are characterized by the temporal dependency of neuronal activation patterns of anatomically separated brain regions (functional connectivity) within networks[16,17]. To this effect, resting-state BOLD-fMRI offers exquisite sensitivity in mapping the effects of drugs on functional connections of specific brain regions, within local networks, and on the global organization of functional communication in the brain[18].

Studies examining the effects of oxytocin on human brain functional connectivity at rest have been scarce and the evidence gathered so far has been rather inconsistent[9]. Exceptions are some effects on the attentional and salience large-scale-networks[19,20]. Overall, this set of studies seem to converge on suggesting that the regulatory role of oxytocin in the human brain is deeply rooted in its modulatory effects on the intrinsic communication between limbic and basal ganglia regions with upstream cortical and downstream cerebellar nodes[9,21,22]. These studies have mostly applied seed-based analysis, a highly recommended approach when a priori hypotheses for the involvement of specific regions/pathways can be formulated, or data-driven approaches such as independent component analysis. The former approach faces the important limitation of not being able to reveal effects on non-hypothesised pathways, which is a real concern in human oxytocin research given the lack of selective radioligands to map the spread and density of oxytocin receptors to inform the selection of seeds; whereas the latter approach does not take into full consideration the complex topological architecture of human brain networks, including information segregation and integration, which have been suggested as pivotal elements of the biological mechanisms underlying behaviour and cognition[23].

The quantitative analysis of complex networks, largely based on graph theory, has introduced new opportunities for understanding the brain as a complex system of interacting elements[16,17], addressing some of the limitation of the analytical approaches outlined above. Thanks to this framework, we have come to appreciate that the human brain relies on fundamental aspects of network organization such as small-world topology, highly connected hubs and modularity[17]. Graph theory conceptualizes the brain as a complex network graphically represented by a collection of nodes and edges. In this virtual graph, nodes constitute anatomical elements (e.g. brain regions), and edges represent the relationships between nodes (e.g., functional or structural connectivity)[17]. Modelling this network allows for the calculation of several metrics of brain topography[16] that can be reliably measured within the same individual[24]. These small-world metrics have been found to predict interindividual differences in cognition[25], change throughout brain development[26] and be sensitive to the effects of aging[27], pathological states[23] or pharmacological compounds[28–30]. In contrast to other widely used resting-state fMRI analytical methods, graph-based network analysis allows us to quantitatively characterize the global and regional organization of the brain[16] and its topological reconfigurations in response to a challenge, which may be an external-task[31] or the administration of a drug[28–30]. Hence, the application of graph theory to the study of the pharmacodynamics of oxytocin in the human brain has the potential to unravel new mechanisms through which oxytocin might modulate human cognition and behaviour. Yet, we are not aware of any study investigating the effects of exogenous oxytocin on the human functional connectome at rest through the lens of graph theory.

In this exploratory study, we administered 40 IU intranasally (using either a standard nasal spray or a nebuliser) or 10 IU intravenously to 16 healthy men using a placebo-controlled, double-blind, triple-dummy, crossover design. We acquired multi-echo resting-state BOLD-fMRI data and used graph-theory modelling, adopting a data-driven approach, to explore the changes in the global and regional topography of the human functional connectome that follow the administration of exogenous oxytocin through both intranasal and intravenous routes, compared to placebo. While standard nasal spray is the predominant method of oxytocin administration in humans, there has been an intense debate about whether the effects of intranasal oxytocin on humans reflect privileged nose-to-brain delivery or result from concomitant increases in systemic oxytocin circulation[32]. Our own work comparing the effects of intranasal and intravenous oxytocin on regional cerebral blood flow in the human brain over an extended period of time suggested that while some of the effects of intranasal oxytocin can be explained by increases in plasmatic oxytocin, other effects seem specific to the intranasal route[11]. The inclusion of an intravenous comparator in the current study allowed us to uncover the effects on the functional connectome related to the presence of oxytocin in systemic circulation. Furthermore, it allowed us to investigate whether our previous conclusions related to the contribution of nose-to-brain pathways to the effects of intranasal oxytocin on brain physiology would be supported by other functional imaging modalities, such as resting-state fMRI. Alongside with the standard nasal spray, we also used a nasal administration method (PARI SINUS nebuliser) that combines the production of small size droplets with vibration to maximize deposition in the olfactory and respiratory epithelia[33] that are thought to mediate the nose-to-brain transport[34]. The inclusion of this second method of intranasal administration allowed us to inspect whether we can maximize the effects of intranasal oxytocin on brain

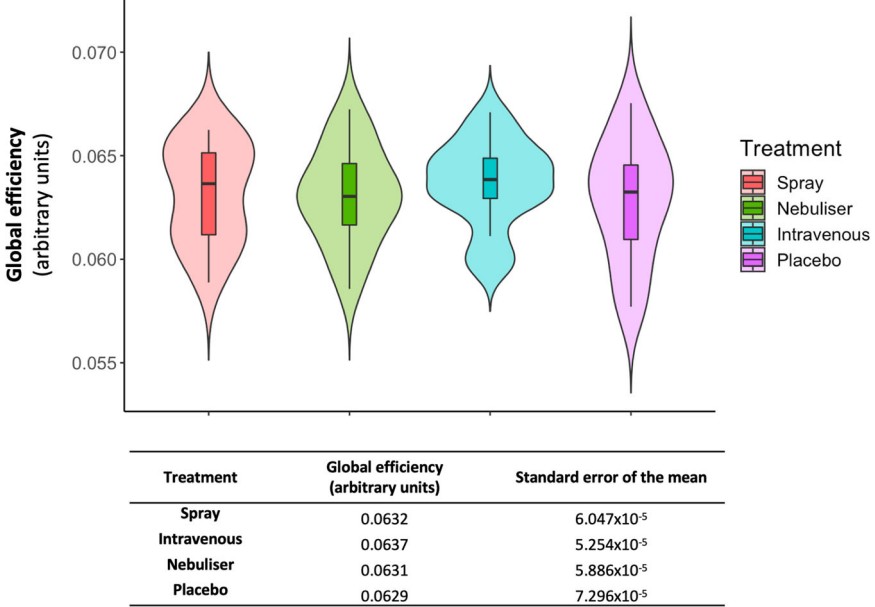

| Treatment | Global efficiency (arbitrary units) | Standard error of the mean |
|---|---|---|
| Spray | 0.0632 | $6.047 \times 10^{-5}$ |
| Intravenous | 0.0637 | $5.254 \times 10^{-5}$ |
| Nebuliser | 0.0631 | $5.886 \times 10^{-5}$ |
| Placebo | 0.0629 | $7.296 \times 10^{-5}$ |

**Fig. 1 Effects of exogenous oxytocin on global efficiency.** We tested the effects of the four treatment conditions on global efficiency using repeated measures one-way analysis of variance. In the upper panel, we present box and violin plots depicting the distribution of the global efficiency measure for each treatment condition; middle horizontal lines represent the median; boxes indicate the 25th and 75th percentiles. In the lower panel, we present the descriptive statistics for each condition ($n = 16$ per treatment condition).

connectivity by increasing its deposition in putative areas of transport to the brain.

In our last perfusion study, we investigated oxytocin induced effects on local perfusion in the brain[11], which do not take into account interactions between brain regions, a core feature of the biology of the brain[17]. Hence, in this study we sought to expand our previous perfusion findings and apply connectomic analyses to investigate how exogenous oxytocin, when administered using different routes and methods for intranasal administration, impacts on the functional connectome. Rather than testing specific hypotheses, we conducted exploratory connectomic analyses at the whole-brain, as an attempt to unravel new aspects of the effects of oxytocin on the brain that previous studies might have missed by simply focusing on key hubs of the brain oxytocin system, such as the amygdala.

Here, we demonstrate that oxytocin modulates the local topography of the human functional connectome at rest within discrete neural systems, but does not affect the global capacity for information transfer among nodes of the functional connectome. We show that in addition to brain systems previously identified as key hubs of the brain oxytocin system (e.g. the amygdala), oxytocin also modulates several brain systems that have not been thoroughly investigated in the field, such as the visual and language circuits, the temporal lobe and the cerebellum. We also demonstrate that the presence of oxytocin in systemic circulation has broad effects on human brain regional connectivity. Yet we also provide indirect evidence supporting the involvement of nose-to-brain pathways in at least some of changes in brain function observed after intranasal oxytocin.

## Results
**Effects of oxytocin on global efficiency and mean functional connectivity (macroscale).** There were no significant differences on the global efficiency (F $(1.932, 28.980) = 0.568$, $p = 0.567$) (Fig. 1) or mean functional connectivity (F$(2.271, 34.070) = 0.774$, $p = 0.484$) across the four treatment conditions (Supplementary Fig. S1).

## Effects of oxytocin on nodal metrics (mesoscale)
*Betweenness-centrality.* We found significant differences between the four treatment conditions on the betweenness-centrality of several nodes spread across the brain (Supplementary Table S1). Post-hoc investigations of this effect allowed us to isolate patterns of changes in the betweenness-centrality of these nodes for each of the three methods of administration tested here. Compared to placebo, oxytocin administered with the spray increased the betweenness-centrality of the right orbitofrontal cortex, left lateral occipital cortex, right temporal occipital fusiform cortex and left occipital pole, while it decreased the betweenness-centrality of the precuneus, right posterior supramarginal and middle temporal gyri, the left anterior middle temporal gyrus and the right cerebellum (area 9) (Fig. 2A and Supplementary Table S1). When administered with the nebuliser, oxytocin increased the betweenness-centrality of the temporo-occipital area of the left middle temporal gyrus and the left cerebellum (area 7) (Fig. 2B and Supplementary Table S2). When administered intravenously, oxytocin increased the betweenness-centrality of the right and left occipital poles, left postcentral gyrus, right supramarginal gyrus and left and right cerebellum (areas 4–5 and area 10, respectively), while it decreased the betweenness-centrality of the right amygdala and the anterior parahipoccampal gyrus (Fig. 2C and Supplementary Table S1). Increases in the betweenness-centrality in the left lateral occipital cortex and the right posterior medial temporal gyrus observed after the administration of oxytocin by spray remained significant when compared directly to the intravenous condition (Supplementary Table S1). Increases in betweenness-centrality in the left temporo-occipital middle temporal gyrus after administration of oxytocin with the nebuliser also remained significant when compared directly to the intravenous condition (Supplementary Table S1).

*Local efficiency.* Our analyses on local efficiency also unveiled differences between the four treatment conditions for several nodes spread across the brain (Supplementary Table S2). A post-hoc investigation of these differences showed that, compared to placebo, oxytocin administered with the spray increased the local

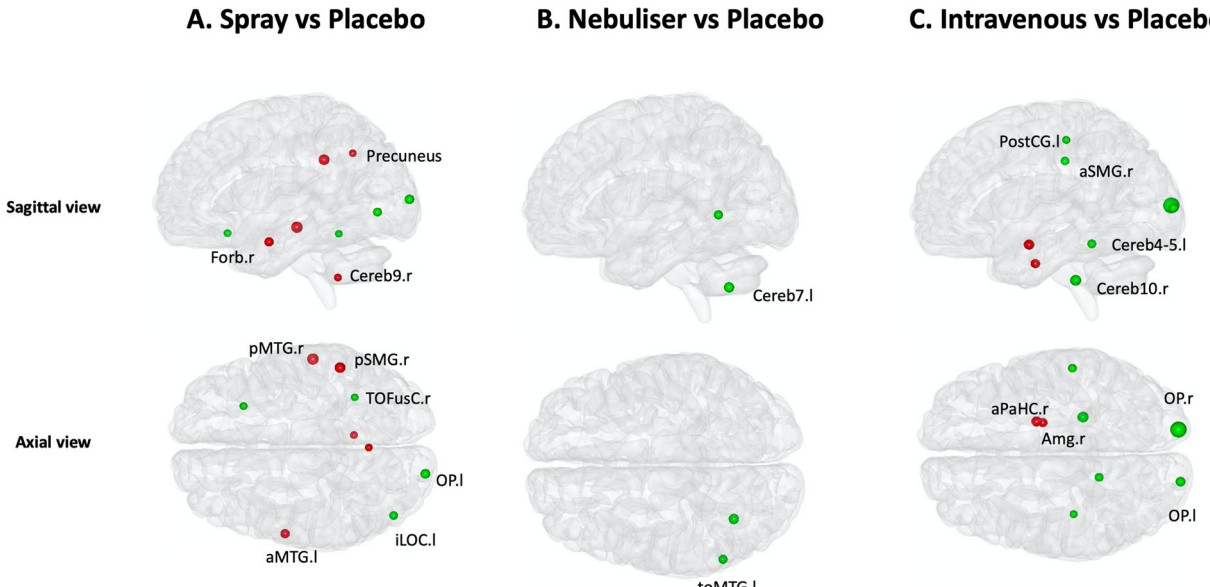

**Fig. 2 Effects of exogenous oxytocin on nodal betweenness-centrality.** For each node, we compared betweenness-centrality between the four treatment conditions using repeated measures one-way analysis of variance. Since we performed tests for multiple nodes, we controlled false positives using FDR correction for the number of nodes examined. When a significant effect was found, we explored this effect further using paired *t*-tests for each pair of treatment groups, correcting for multiple testing using FDR. In this figure, we provide an overview of the pattern of changes in betweenness-centrality for each method of administration by showing the results of our post-hoc tests focusing on the comparisons between each active treatment arm and placebo (for direct comparisons between active treatment arms, please refer to Table S1). The green and red colours depict increases and decreases in betweenness-centrality, respectively. The size of each nodal sphere is proportional to the T-statistic of each comparison. The name of each node appears once, in the axial or sagittal view, to avoid cluttering the figure. R right, l left, a anterior, p posterior, Forb orbitofrontal cortex, MTG medial temporal gyrus, SMG supramarginal gyrus, TOFusC temporal occipital fusiform cortex, OP occipital pole, iLOC lateral occipital cortex, Cereb Cerebellum; PostCG Postcentral gyrus, Amg Amygdala, aPaHC anterior parahippocampal gyrus, toMTG temporo-occipital area of the middle temporal gyrus.

efficiency of the right superior frontal gyrus, right anterior and posterior medial temporal gyri, right posterior superior temporal gyrus, right cuneal cortex and left cerebellum (area 1) (Fig. 3A and Supplementary Table S2). When administered with the nebuliser, oxytocin increased the local efficiency of the right cuneal cortex and the left posterior supramarginal gyrus (Fig. 3B and Supplementary Table S2). Administered intravenously, oxytocin increased the local efficiency of the right superior frontal gyrus, right frontal pole, anterior cingulate, left cuneal cortex, right posterior supramarginal gyrus and the brainstem (Fig. 3C and Supplementary Table S2). The increases in the right posterior medial temporal gyri observed after the spray were still significant when compared directly with the intravenous condition (Supplementary Table S2).

*Node degree.* To a lesser extent, we found significant differences between the four treatment conditions for node degree, which were restricted to the posterior brain (Supplementary Table S3). A post-hoc investigation of these differences showed that, compared to placebo, oxytocin administered with the spray increased node degree of the brainstem, left cerebellum (area 1) and right intracalcarine cortex, while it decreased the node degree of the right posterior inferior temporal gyrus (Fig. 4A and Supplementary Table S3). When administered with the nebuliser, it increased the node degree of the left posterior superior temporal gyrus and left cerebellum (area 7) (Fig. 4B and Supplementary Table S3). When administered intravenously, it decreased the node degree of the precuneus and right temporal pole (Fig. 4C and Supplementary Table S3). This increase in the left posterior superior temporal gyrus after the administration with the nebuliser was still significant when compared directly to the intravenous condition (Supplementary Table S3).

**Effects of oxytocin on network edges (microscale).** We did not find any individual edge where the four treatment conditions differed in connectivity after applying the stringent FDR correction for the number of edges. Repeating this analysis with NBS also did not identify any subnetwork where the four treatment conditions differed on connectivity after correcting for multiple testing, irrespective of the primary threshold used.

**Functional decoding.** The functional decoding of the pathways for which we identified significant modulatory effects of oxytocin revealed terms mostly associated with early visual processing, mentalizing and theory-of-mind, social and audiovisual processing, facial processing, expectancy, aversion, somatosensation (pressure and nociception) and heart physiology (Fig. 5).

**Overlap with large-scale resting-state networks.** The pathways for which we identified significant modulatory effects of oxytocin (Fig. 6A) overlapped primarily with regions belonging to the visual network (Dice coefficient = 0.68), followed by the frontoparietal (Dice coefficient = 0.54), default-mode (Dice coefficient = 0.49), limbic (Dice coefficient = 0.48) and ventral attention networks (Dice coefficient=0.44). The overlap with the dorsal attention (Dice coefficient = 0.20) and sensorimotor (Dice coefficient = 0.24) networks was considerably lower (Fig. 6B).

**Effects of exogenous oxytocin administration on plasma oxytocin concentration at 57 min post-dosing.** We found a significant interaction Time × Treatment in plasma oxytocin concentration (F (3, 46.686) = 25.550, p < 0.0001). Post-hoc explorations revealed that, at baseline, the four treatment

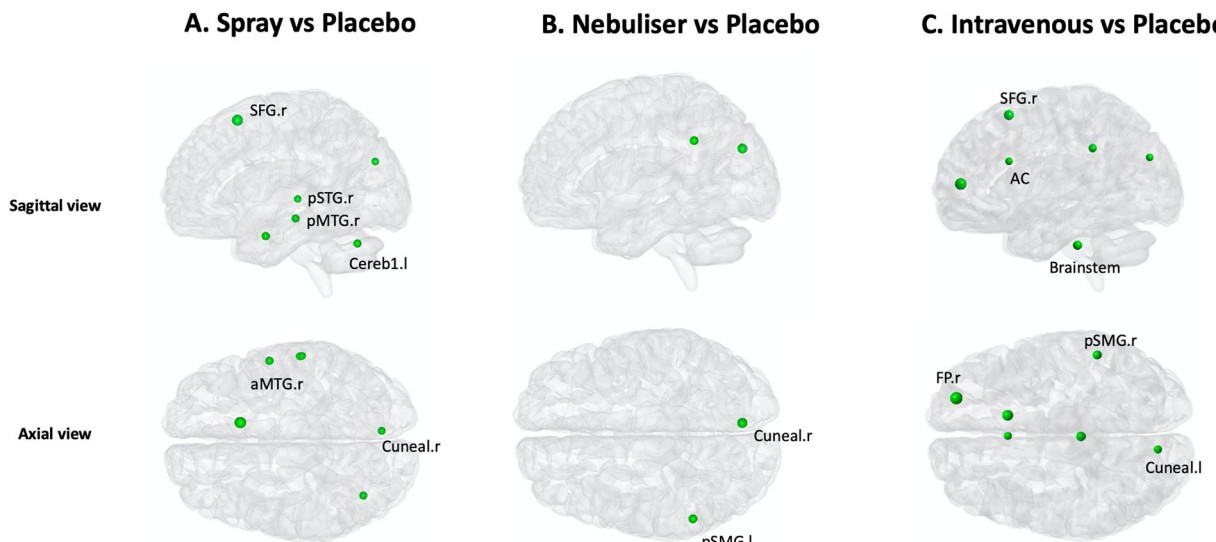

**Fig. 3 Effects of exogenous oxytocin on nodal local efficiency.** For each node, we compared local efficiency between the four treatment conditions using repeated measures one-way analysis of variance. Since we performed tests for multiple nodes, we controlled false positives using FDR correction for the number of nodes examined. When a significant effect was found, we explored this effect further using paired t-tests for each pair of treatment groups, correcting for multiple testing using FDR. In this figure, we provide an overview of the pattern of changes in local efficiency for each method of administration by showing the results of the post-hoc tests focusing on the comparisons between each active treatment arm and placebo (for direct comparisons between active treatment arms, please refer to Table S2). The green and red colours depict increases and decreases in local efficiency, respectively. The size of each nodal sphere is proportional to the T-statistic of each comparison. The name of each node appears once, in the axial or sagittal view, to avoid cluttering the figure. R right, l left, a anterior, p posterior, SFG superior frontal gyrus, STG superior temporal gyrus, MTG medial temporal gyrus, SMG supramarginal gyrus, Cereb Cerebellum, Cuneal cuneal cortex, AC anterior cingulate, FP frontal pole.

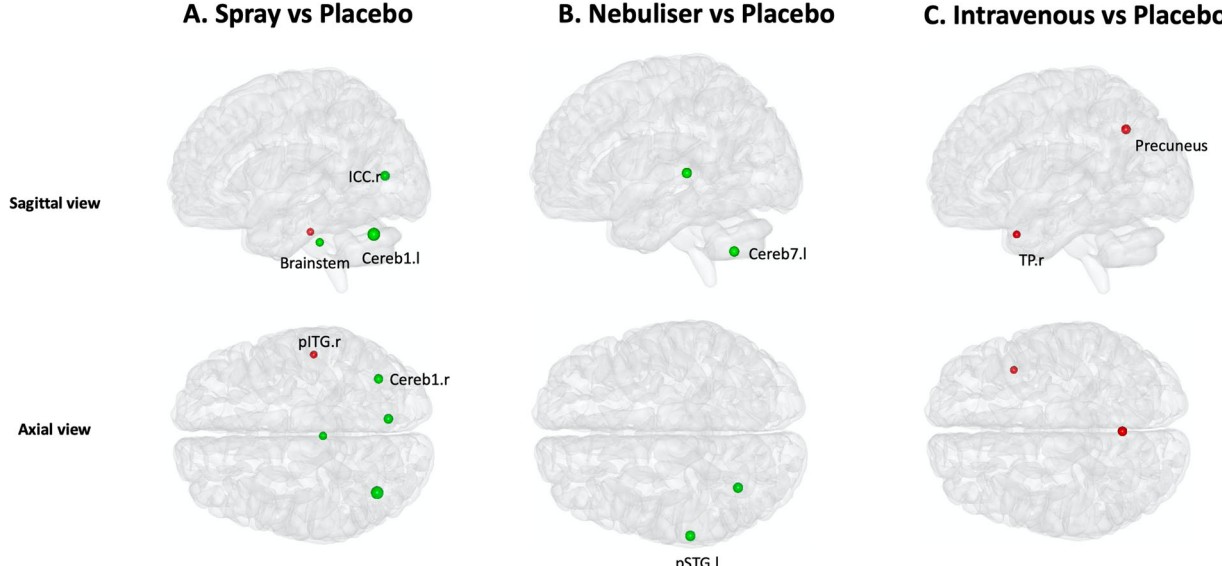

**Fig. 4 Effects of exogenous oxytocin on nodal degree.** For each node, we compared node degree between the four treatment conditions using repeated measures one-way analysis of variance. Since we performed tests for multiple nodes, we controlled false positives using FDR correction for the number of nodes examined. When a significant effect was found, we explored this effect further using paired t-tests for each pair of treatment groups, correcting for multiple testing using FDR. In this figure, we provide an overview of the pattern of changes in node degree for each method of administration by showing the results of the post-hoc tests focusing on the comparisons between each active treatment arm and placebo (for direct comparisons between active treatment arms, please refer to Table S3). The green and red colours depict increases and decreases in node degree, respectively. The size of each nodal sphere is proportional to the T-statistic of each comparison. The name of each node appears once, in the axial or sagittal view, to avoid cluttering the figure. R right, l left, a anterior, p posterior, ICC intracalcarine cortex, Cereb Cerebellum, ITG inferior temporal gyrus, TP temporal pole, STG superior temporal gyrus.

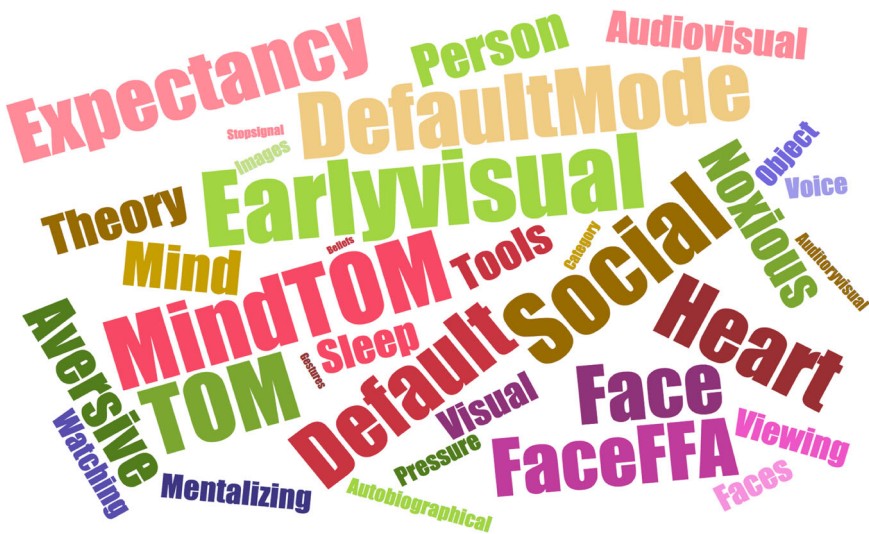

**Fig. 5 Functional characterization of the neural circuits engaged by exogenous oxytocin at rest in healthy men.** The word cloud presents the top terms derived from the NeuroSynth decoder for the neural pathways engaged by oxytocin using reverse inference, after excluding terms for brain regions or imaging methods. Font size is coded by the correlation strength between the pathway and term.

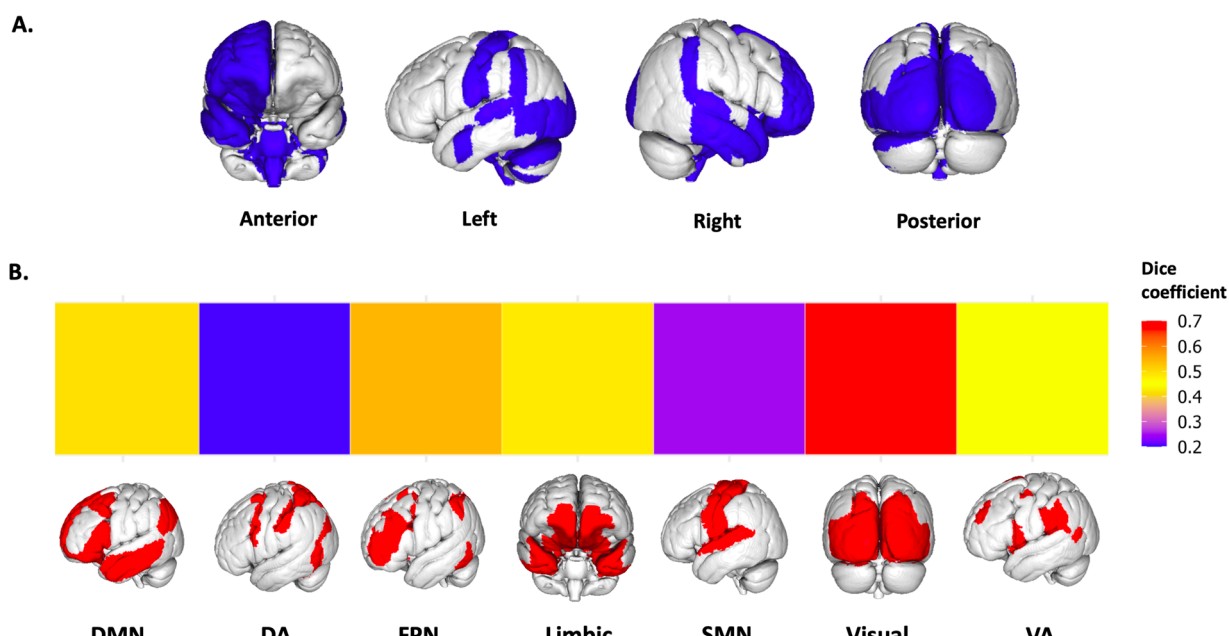

**Fig. 6 Overlap between the neural circuits engaged by exogenous oxytocin at rest and the canonical large-scale resting-state networfks.** We calculated the percentage of overlap between our single binary map including all regions showing alterations in regional functional connectivity after oxytocin (irrespective of route/method of administration) and the large-scale resting-state networks described in the atlas from Yeo et al. (2011)[105]. Overlap was quantified using the Dice coefficient, which measures the percentage of voxels of each resting-state network overlapping with our treatment effect mask. In the upper panel A, we provide an overview of all regions where we found treatment effects, irrespective of route/method of administration, rendered in a 3D surface model. In the lower panel B, we provide a heatmap summarizing the percentage of overlap (Dice coefficient) between the regions in A and each of the seven networks (each network rendered in a 3D surface model). DMN Default-mode network, DA Dorsal attention, FPN Frontoparietal network, SMN Sensorimotor network, VA Ventral attention.

conditions did not differ in plasma oxytocin concentration (minimal $p_{Tukey} = 0.540$). Intranasal and intravenous administrations of oxytocin resulted in increased plasma oxytocin concentration at 57 min post-administration, compared to baseline (all $p_{Tukey} < 0.010$), but the concentration of plasma OT in the placebo condition did not change significantly between the two timepoints ($p = 0.460$). Plasmatic oxytocin concentrations at

post-administration were higher in the intravenous condition than in all the other three treatment conditions (all $p_{Tukey} < 0.010$). Spray and nebuliser did not differ in plasma oxytocin concentrations at 57 min post-administration ($p_{Tukey} = 1.000$) (Fig. 7). These results are consistent with the pharmacokinetics of exogenous oxytocin for the full post-dosing interval (15–104 min post-dosing) reported in a previous study[11].

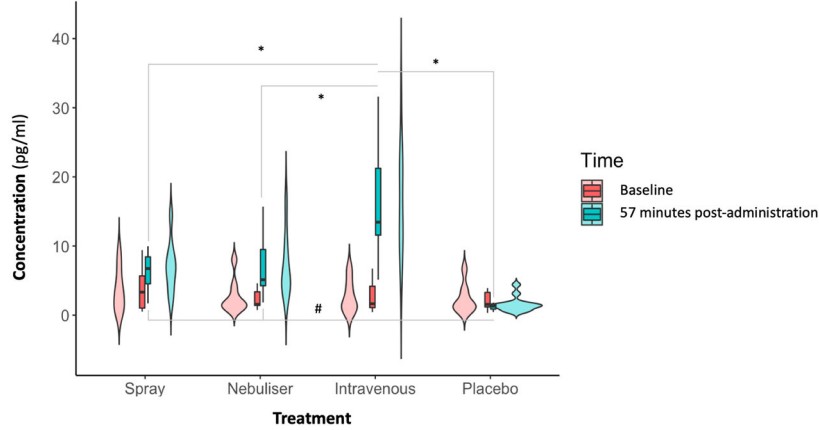

| Time | Treatment | Concentration (pg/ml) | Standard error of the mean |
|---|---|---|---|
| | Spray | 3.701 | 0.757 |
| Baseline | Nebuliser | 2.689 | 0.589 |
| | Intravenous | 2.763 | 0.565 |
| | Placebo | 2.345 | 0.528 |
| | Spray | 7.034 | 0.929 |
| 57 minutes | Nebuliser | 7.207 | 1.151 |
| post-administration | Intravenous | 16.35 | 1.802 |
| | Placebo | 1.577 | 0.234 |

**Fig. 7 Baseline and post-administration (57 min from last treatment offset) concentrations of plasmatic oxytocin in each treatment condition.** We conducted a linear mixed model on plasmatic oxytocin concentrations measured in a blood sample collected immediately before the beginning of the resting-state acquisition, using Time, Treatment and Time × Treatment as fixed factors and a random intercept for subject. We followed up the significant interaction with further post-hoc tests to examine differences between each pair of treatment conditions between baseline and post-administration and between treatment conditions within each time level, using Tukey's correction for multiple testing. In the upper panel, we present box and violin plots depicting the distribution of plasmatic concentrations of oxytocin for each treatment condition and time point; middle horizontal lines represent the mean; boxes indicate the 25th and 75th percentiles ($n = 16$ per treatment condition and time point). In the lower panel, we present the descriptive statistics for each level of the two factors. *Intravenous condition compared to the other three ($p < 0.01$) at post-administration; # Spray and Nebuliser conditions compared to placebo (<0.001) at post-administration.

## Discussion

Motivated by the current lack of clarity regarding the central targets of exogenous oxytocin, we conducted an exploratory data-driven examination of the effects of exogenous oxytocin on the human brain functional connectome at rest using different routes of administration. By identifying the functional substrates that oxytocin engages at rest we expected to provide insights that help understand its effects on human brain and behaviour. We discuss each of our main findings below.

Our first key finding was the observation that exogenous oxytocin, compared to placebo, irrespective of method of administration, modulates the regional topography of the human functional connectome at rest for a range of regions spanning the temporal lobes, the occipital, parietal and frontal cortices, the amygdala, the anterior cingulate cortex, the brainstem and the cerebellum. In contrast, the lack of effects on the macroscale measures (global efficiency and mean functional connectivity) suggests that oxytocin does not modulate the global capacity for information transfer among the nodes of the functional connectome. The lack of effects on a global scale is compatible with the assumed discrete distribution of the receptors for which oxytocin has affinity in the human brain (i.e. the oxytocin receptor, and to a lesser extent, the vasopressin receptor)[35]. It is also consistent with a large number of reports, mostly from rodents, which have demonstrated neuromodulatory effects of oxytocin at the circuit level in specific brain regions (for an extensive review see refs. [36,37]).

At first glance, the regional effects we found seem dispersed across many brain regions, without a very clear organizational pattern. Nevertheless, a careful examination of these changes suggests that some of these regions are part of well-described neuronal circuits that operate together in a number of brain

functions. Indeed, the brain pathways we found to be modulated by exogenous oxytocin overlapped considerably with a number of well-described canonical large-scale resting-state networks, namely the visual, frontoparietal, default-mode, limbic and ventral attention networks. We note that two previous studies applying hypothesis-driven analyses have reported consistent effects of a single dose of intranasal oxytocin on the default-mode, salience and attention networks, including on their between-network connectivity (other networks were simply not investigated)[19,20]. Hence, while our data is broadly compatible with these two studies, we expand their conclusions in, at least, one important direction. We show that most likely the effects of oxytocin on brain processing involve several brain networks and can hardly be attributed to a single neural pathway. Instead, oxytocin most likely operates at multiple levels, spanning basic sensory processing to salience detection and attention orientation, emotional processing and high-order mental representations of the self and others.

Since our data was acquired at rest, inferences on the behavioural relevance of the observed effects should be made with caution. However, it seems that, overall, oxytocin can prime, at rest, neural circuits that have been described to contribute to a number of mental/behavioural processes—that we summarized through functional decoding using the Neurosynth database. These processes were mostly related to early visual processing, mentalizing and theory-of-mind, social and audiovisual processing, facial processing, expectancy, aversion, somatosensation (pressure and nociception) and heart physiology. Putting all of these processes together, we suggest that an overarching model for the effects of oxytocin on human behaviour could be summarized in the following three main mechanisms: (1) oxytocin may facilitate the processing of socially relevant sensory cues and

enhance our capacity to create cognitive models of others' mental and affective states, which together may concur to adjust behaviour according to the dynamic interplay between sender (others) and receiver (self)[38]; (2) oxytocin may facilitate approach behaviour towards others, by enhancing the motivational value of social interactions and decreasing approach avoidance/anxiety[39]; (3) oxytocin might facilitate the integration of cognitive processing with physiological signals, contribute to the encoding of the saliency or precision of interoceptive signals, and facilitate the acquisition of generative brain models of the emotional and social selves[40].

Studies investigating the effects of oxytocin on resting-state functional connectivity have been limited and restricted to intranasal administrations using the standard spray (for an overview please see refs. [9,21,22]). The vast majority of these studies examined changes in either amygdala/basal ganglia connectivity after intranasal oxytocin and highlighted that intranasal oxytocin may exert its behavioural and social cognitive effects by altering the connectivity between regions responsible for social and emotive processing (i.e. the amygdala, medial prefrontal cortex, anterior cingulate cortex, basal ganglia, precuneus/posterior cingulate cortex). While consistent with these reports, our findings show that the effects of exogenous oxytocin on human brain functional connectivity may go beyond those areas typically inspected/reported in the literature and that they encompass many other regions, namely temporal and occipital areas, the cerebellum and the brainstem. We note that we did not find any effect of exogenous oxytocin on the regional connectivity of the basal ganglia, despite the well-known presence of the oxytocin receptor in this area[35] and previous reports showing modulation of the intrinsic connectivity at rest between these and other areas of the brain at about 40 mins post-dosing[21,22]. Considering that our data was acquired about 17 min later, it is conceivable that the effects of oxytocin on the functional connectivity of the basal ganglia happened at a post-dosing window that we did not sample.

Our second key finding was the observation that the regional effects of exogenous oxytocin, compared to placebo, vary as a function of the route/method of administration. Particularly striking was the observation that the intravenous administration of oxytocin induced widespread changes in regional functional connectivity across many areas potentially relevant to explaining the effects of intranasal oxytocin on human behaviour and which have been classically ascribed to direct nose-to-brain transport. One of these areas is the amygdala, where we found decreased betweenness-centrality after intravenous oxytocin compared to placebo, an effect that might reflect a reduced dominance of this area in transmitting information. This result is also consistent with the dampening of the amygdala, which has been consistently demonstrated in animal studies and intranasal oxytocin studies in men[41–44]. We have previously shown that the same dose of intravenous oxytocin we used here decreases rCBF in the amygdala/parahippocampal gyrus and in the anterior cingulate in the same cohort of participants[11]. Our current findings not only confirm, using another functional imaging modality, the ability of intravenous oxytocin to target these areas, but also show that the effects of oxytocin when administered intravenously may be broader and encompass other cortical areas, including the postcentral gyrus, frontal, temporal and occipital areas, the brainstem and the cerebellum. We can therefore infer that the presence of oxytocin in systemic circulation can influence brain function in a wide range of circuits potentially related to the behavioural effects associated with intranasal oxytocin. For example, our data showed increases in the betweenness-centrality of the right occipital pole and increases in local efficiency in the right superior frontal gyrus, both for the intranasal (by spray) and the

intravenous administration of oxytocin, when compared to placebo; direct comparisons between spray and intravenous administrations for these nodes did not yield any significant differences. Therefore, our findings call for caution in attributing all of the neural/behavioural effects of intranasal oxytocin to direct nose-to-brain pathways.

It is not yet clear which precise mechanisms mediate the effects of intravenous oxytocin on brain function. Oxytocin is a hydrophilic peptide that only crosses the blood–brain barrier (BBB) in very small amounts[45]. At least three mechanisms have been postulated. First, it is possible that the direct peripheral effects of oxytocin on oxytocin receptors expressed in vegetative territories[46], such as the heart, may be an indirect source of changes in the brain. However, we have previously shown the absence of effects on heart-rate or heart-rate variability for the same doses/methods and cohort of participants we used here[11], which makes the contribution of changes in heart physiology unlikely. Yet, we cannot discard the hypothesis that effects on other peripheral organs may still contribute to the effects on connectivity reported here. Second, it is possible that the small amounts of oxytocin that cross the BBB[45] (or a metabolite of oxytocin that remains functional) are sufficient to induce changes in brain connectivity[47]. Third, and related to the above, it is possible that the small amount of synthetic oxytocin crossing the BBB is sufficient to engage oxytocin autoreceptors on the oxytocin-synthesising neurons in the hypothalamus, inducing the release of endogenous oxytocin in the brain in a positive feedback loop mechanism[48]. From these three hypotheses, the second one has received perhaps the most robust experimental evidence until now, with one study reporting that the intravenous infusion of labelled synthetic oxytocin increased synthetic oxytocin levels in the CSF in primates[49], and another one showing that circulating oxytocin can be transported into the brain by the receptors for advanced glycation end-products (RAGE) on brain capillary endothelial cells[50].

While we show that systemic oxytocin produces widespread changes in the functional connectome at rest and that at least some of these effects mirror the changes observed after the intranasal administration of exogenous oxytocin, we also show that the presence of oxytocin in systemic circulation does not fully account for all the changes we identified following the intranasal administration of oxytocin, compared to placebo. For instance, increases in betweenness-centrality in the left lateral occipital cortex or the right posterior medial temporal gyrus observed after the administration by spray were not observed after intravenous administration compared to placebo. Additionally, the direct comparison of these two methods confirmed that these effects were specific for the intranasal spray route. Other examples include the increases in betweenness-centrality in the left temporo-occipital middle temporal gyrus and in node degree in the left superior temporal gyrus after the administration of oxytocin with the nebuliser compared to placebo. These effects were not observed after the intravenous oxytocin administration (compared to placebo) and remained significant when we directly compared these two methods of administration. There were many other effects induced by the intranasal (either by spray or nebuliser) administration of oxytocin, compared to placebo, which were not seen following intravenous oxytocin administration, and vice-versa. However, in these cases the direct comparisons between the intranasal and intravenous routes yielded no differences. Hence, for these effects we cannot be certain about the exact route they originated from.

Our findings support our previous observations that increases in oxytocin in systemic circulation cannot fully account for the changes in rCBF observed after intranasal oxytocin[11] and lend support to the contribution of nose-to-brain pathways to the

effects of intranasal oxytocin on brain function[51]. However, they cannot highlight the precise mechanisms of transport underlying its functional effects. We believe that most evidence to date concurs on the idea that oxytocin, when administered intranasally, may diffuse from the olfactory and respiratory epithelia in the middle and upper posterior regions of the nasal cavity along ensheathed channels surrounding the olfactory and trigeminal nerve fibre pathways to the CSF and/or the brain[52]. This hypothesis is supported by animal studies[53–57] and fits the temporal dynamics of at least some of the functional effects of intranasal oxytocin on the human brain. In line with this hypothesis, one recent study on rhesus monkeys showed that, when administered intranasally, labelled synthetic oxytocin reaches several brain regions (orbitofrontal cortex, striatum, thalamus and brainstem) positioned in the trajectories of the olfactory and trigeminal nerves. No detectable amounts of labelled oxytocin were observed in these brain regions when the same doses were administered intravenously[58].

Our third key finding was the observation that while the application of the same nominal dose of intranasal oxytocin (40 IU) with the standard nasal spray and the nebuliser resulted in identical pharmacokinetic profiles, the patterns of changes in regional connectivity were markedly different across the two methods of intranasal administration. Indeed, excluding the increases in the local efficiency observed in the right cuneal cortex for both the spray and the nebuliser, we could not find any other instances where these two methods converged. These differences are consistent with our previous findings on rCBF, where we also reported considerable differences in the patterns of changes in perfusion achieved by each method despite their similar pharmacokinetic profiles[11]. We hypothesize that these differences between the two intranasal methods are likely to be explained by higher oxytocin deposition by the nebuliser[34] in the olfactory and respiratory regions, and the parasinusal cavities, resulting in increased amounts of oxytocin reaching the brain (compared to the standard nasal spray).

At first glance, our findings may be difficult to reconcile with the hypothesis outlined above. If the nebuliser is more efficient in delivering oxytocin centrally, one would intuitively expect this method to produce stronger effects in areas where the same dose, when administered by spray, resulted in significant modulations, or even produce effects where the same nominal dose of oxytocin, when administered by spray, is insufficient. However, this prediction would mostly be valid if the pharmacodynamics of oxytocin in the brain followed a linear model—which is not the case, at least for some brain areas. Indeed, the few studies inspecting the dose-response effects of intranasal oxytocin, for example, on amygdala reactivity, support an inverted-U shape dose-response curve by showing that deviating from an "optimal" dose may in fact result in lower or null effects[59,60]. Our hypothesis invites for future dose-response studies using the nebuliser to inspect whether lowering the dose administered with the nebuliser may approximate the effects on the functional connectome achieved after the administration of a high dose of oxytocin with the standard spray.

Our study faces certain limitations that we should acknowledge. First, while our study provides proof-of-concept of the potential of connectomics as a new approach to study the pharmacodynamics of exogenous oxytocin in the human brain, our sample size was not optimized to capture effects of oxytocin on the connectome smaller than a medium effect size. Hence, it is possible we might have missed smaller effects for at least some of the different dependent variables examined here (this limitation may have particularly affected our analysis on individual edges, where we applied stringent control for multiple testing across the number of edges). There has been discussion about the minimal sample size

and amount of data required to estimate graph metrics of functional connectivity with sufficient robustness[61,62]. While some studies provided evidence that these metrics can be reliably measured across a wide range of sample sizes ($5 < n < 45$)[62], another study has suggested that only sample sizes in the order of 100 individuals (40 individuals if combined with long fMRI acquisitions) are enough to estimate graph metrics with sufficient reliability[63]. However, this study used a suboptimal denoising pipeline, which could have hindered the reliability of the estimated graph metrics. Indeed, studies have shown that different methodological options related to, for instance, the type of cleaning of BOLD signal used and the type of connectivity matrices used to estimate the graphs (weighted versus unweighted) can affect the reliability of these metrics. In this study, we attempted to maximize the reliability of the graph metrics we used by acquiring multi-echo data and applying a denoising pipeline that has been shown to provide better signal-to-noise ratio as compared to single-echo acquisitions or multi-echo acquisitions denoised with other suboptimal approaches[64], i.e. a standard regression of motion parameters and WM and CSF signals. Moreover, we relied on weighted graphs to further maximize reliability. Furthermore, we also estimated the graph metrics using the areas under the curve of each metric across different costs, which provides a more robust estimate of each metric independently of specific choices for the cost used to build the graphs. Further work needs to be done to investigate the impact of methodological choices that improve data quality on the reliability of network-based metrics, such as those we used here. Irrespectively of this, future studies should attempt to replicate our approach with larger samples, preferentially applying a dose-response design and sampling different intervals post-dosing. Second, we applied FDR to correct for multiple statistical tests. This approach provides a principled means for containing the rate of false positives within acceptable limits. However, univariate tests are not ideal for connectivity analysis due the multivariate nature of the data. An alternative would be to compare patterns of connectivity while accounting for covariance within the data[28] using pattern recognition. Nevertheless, our sample size would be far too small to conduct such analysis. Third, our findings cannot be readily extrapolated to women, given the known sexual dimorphism of the oxytocin system in the brain and behavioural responses to oxytocin in humans[65–67]. Fourth, the amount of oxytocin we administered intravenously was not chosen to mimic exactly the plasmatic concentrations achieved after intranasal administration, which would require the infusion of a dose about five times lower (2IU)[60]. Instead, we adopted a proof-of-concept approach, aiming to achieve consistently higher plasmatic concentrations of oxytocin during the full period of scanning, eliminating the hypothesis that negative findings could be ascribed to insufficient dosing. Future studies should compare the effects of different doses administered intravenously, including those mirroring the exact increases in plasmatic oxytocin produced by intranasal methods of administration. Last, while we are not aware of any study reporting expression of oxytocin receptors in the vasculature system in the brain, oxytocin may have effects on the BOLD signal that are mediated by oxytocin receptors on the small arterial walls of the cerebral vasculature rather than by neuronal receptors. This is a general limitation of pharmacological studies using fMRI[68]. Further studies looking at effects of oxytocin on vascular reactivity during hypercapnic challenges[69] may uncover whether this should be a concern in BOLD-fMRI studies with oxytocin or not.

In summary, using an exploratory connectomics approach, we demonstrate that oxytocin modulates the local topography of the human functional connectome at rest within discrete neural systems, but does not affect the global capacity for information transfer among nodes of the functional connectome. We show

that in addition to brain systems previously identified as key hubs of the brain oxytocin system (e.g. the amygdala), oxytocin modulation involves several brain systems that have not been thoroughly investigated in the field, such as the visual and language circuits, the temporal lobe and the cerebellum, but might contribute to the broad modulatory role of oxytocin on social-emotional behaviour in humans. We demonstrate that the presence of oxytocin in systemic circulation has broad effects on human brain regional connectivity and may account for at least some of the effects on brain function and behaviour typically ascribed to direct nose-to-brain transport after intranasal oxytocin. At the same time, we also provide indirect evidence supporting the involvement of nose-to-brain pathways in at least some of changes in brain function observed after intranasal oxytocin. Together, our results suggest that an overarching model for the role of oxytocin in human behaviour should consider modulation of multiple levels of the brain processing hierarchy relevant for socio-affective related behaviours, from basic sensory processing to high-level theory-of-mind inference.

## Methods

**Participants**. We enrolled 17 healthy male adult volunteers (mean age 24.5, SD = 5, range 19–34 years). One participant did not complete one of the four visits and for this reason was excluded from all analyses. We decided to focus our study on men because the levels of oxytocin have been shown to fluctuate across the menstrual cycle in women, which would introduce an additional level of unwanted variability in our sample[70]. We screened participants for psychiatric conditions using the Symptom Checklist-90-Revised[71] and the Beck Depression Inventory-II[72] questionnaires. Participants were not taking any prescribed drugs, did not have a history of drug abuse and tested negative on a urine panel screening test for recreational drugs. Recreational drugs, such as MDMA, have been shown to increase the release of oxytocin to the plasma[73,74], which could have impacted on our findings. Participants were required to consume <28 units of alcohol per week and <5 cigarettes per day. We further instructed participants to abstain from alcohol and heavy exercise for 24 h before scanning, and to restrain from any beverage or food consumption or smoking (if they were smokers) for 2 h before scanning. Intense physical exercise[75] and food intake[76] have been shown to impact on oxytocin release and could have introduced additional noise. Furthermore, nicotine has robust acute[77] and subacute[78] effects on brain physiology, which could have affected our neuroimaging metrics in unpredictable ways. Only three participants were regular smokers (2–3 cigarettes/day) and one participant used to smoke occasionally (around 5 cigarettes/month) at the time of the study.

Participants gave written informed consent. King's College London Research Ethics Committee (PNM/13/14-163) approved the study. The data presented in this manuscript were acquired in the context of a larger study examining the effects of different routes of administration of exogenous oxytocin on cerebral blood flow (published separately in ref. [11]). Our sample size would allow us to detect a medium effect size $f = 0.25$ with 80% of statistical power in a repeated measures one-way ANOVA testing for differences across our four treatment conditions in this study.

**Study design**. We employed a double-blind, placebo-controlled, triple-dummy, crossover design. Participants visited our centre for 4 experimental sessions spaced 8.90 days apart on average (SD = 5.65, range: 5–28 days). In each session, participants received treatment via all three administration routes, in one of two fixed sequences: either nebuliser/intravenous infusion/standard nasal spray, or standard nasal spray/intravenous infusion/nebuliser, according to the treatment administration scheme presented in Fig. 8 (upper panel). In three out of four sessions only one route of administration contained the active drug; in the fourth session, all routes delivered placebo or saline. Participants were randomly allocated to a treatment order (i.e. a specific plan regarding which route delivered the active drug in each experimental session) that was determined using a Latin square design. Unbeknown to the participants, the first treatment administration method in each session always contained placebo (see Fig. 8, upper panel), while intranasal (spray or nebuliser) oxytocin was only delivered with the third treatment administration. This protocol maintained double-blinding while avoiding the potential washing-out of intranasally deposited oxytocin (as might have been the case if oxytocin had been administered at the first treatment administration point and placebo at the third administration point). All sessions were conducted during the morning to minimise potential circadian variability in resting brain activity[79] or oxytocin levels[80].

**Intranasal administration of oxytocin**. For the intranasal administrations, participants self-administered a nominal dose of 40 IU oxytocin (Syntocinon; 40 IU/ml; Novartis, Basel, Switzerland), one of the highest clinically applicable doses[81]

that would insure that lack of effects on the brain are unlikely to result from under-dosing. We have shown that 40 IU delivered with a standard nasal spray induce robust rCBF changes in two different cohorts of healthy men using between-[12] and within-subjects[11] designs. Work from others have also reported this dose to modulate resting functional connectivity[19,82]. For the intranasal administration, we used specially manufactured placebo that contained the same excipients as Syntocinon except for oxytocin. Participants self-administered intranasal oxytocin or placebo with the nasal spray or nebuliser following under the supervision of the lead experimenter who ensured the correct application of the devices. Additionally, all participants had been trained during the screening visit in the correct administration procedures.

*Nasal spray administration*. Participants followed our standard in-house protocol that we have used consistently across all of our studies and is consistent with the guidelines outlined by Guastella et al. (2013)[83]. Specifically, participants self-administered 10 puffs, each containing 0.1 ml Syntocinon (4IU) or placebo, one puff every 30 s, alternating between nostrils (hence 40 IU OT in total). Participants blocked opposite nostril while administering each puff, and each puff was followed by an immediate, brief, sharp, snort.

*PARI SINUS nebuliser*. Participants self-administered 40 IU oxytocin (Syntocinon) or placebo, by operating the SINUS nebulizer for 3 min in each nostril (6 min in total), according to instructions. The correct application of the device was confirmed by determining gravimetrically the administered volume. Participants were instructed to breathe using only their mouth and to keep a constant breathing rate with their soft palate closed, to minimize delivery to the lungs. Previous studies using this nebuliser[34] have shown up to 9.0% (±1.9%) of the total administered dose to be delivered to the olfactory region, 15.7% (±2.4%) to the upper nose.

**Intravenous administration**. For the intravenous administration, we delivered 10 IU oxytocin (Syntocinon injection formulation, 10 IU/ml, Alliance, UK) or saline via slow infusion over 10 min (1IU/min). A 50-ml syringe was loaded with either 32 ml of 0.9% sodium chloride (placebo) or 30 ml of 0.9% sodium chloride with 2 ml of Syntocinon (10 IU/ml). A Graseby pump was used to administer 16 ml of the compound (hence 10 IU of oxytocin in total) over 10 min, at a rate of 96 ml/h. We selected the intravenous dose and rate of administration to assure high plasmatic concentrations of oxytocin throughout the observation period while restricting cardiovascular effects to tolerable and safe limits. A rate of 1IU per minute is typically used in caesarean sections and is considered to have minimal side effects[84,85].

**Procedure**. Each experimental session began with the treatment administration protocol that lasted about 22 min in total (Fig. 8, lower panel). After drug administration, participants were guided to the MRI scanner, where eight pulsed continuous arterial spin labelling scans (each lasting approx. eight minutes) were acquired, spanning 15–104 min post-dosing (data reported elsewhere[11]). The 8-min resting state BOLD-fMRI scan reported herein was obtained at about 57 ± 3.38 min post-dosing. Participants were instructed to lie still and maintain their gaze on a centrally placed fixation cross during scanning. We assessed participants' levels of alertness (anchors: alert-drowsy) and excitement (anchors: excited-calm) using visual analogue scales (0–100) at three different time-points during the scanning session to evaluate subjective drug effects across time (data reported elsewhere[11]).

**Blood sampling and quantification of plasma oxytocin**. We collected plasma samples at baseline and at five timepoints post-dosing (as detailed in Fig. 8, lower panel) to measure changes in the concentration of oxytocin. Plasmatic oxytocin was assayed by radioimmunoassay (RIAgnosis, Munich, Germany) after extraction[86]. The detection limit of this assay is in the 0.1–0.5 pg/sample range. Cross-reactivity with vasopressin, ring moieties and terminal tripeptides of both oxytocin and vasopressin and a wide variety of peptides comprising 3 (alpha-melanocyte-stimulating hormone) up to 41 (corticotrophin-releasing factor) amino acids are <0.7% throughout. The intra- and interassay variabilities are <10%[11,87]. Here, we focused on the concentrations of plasmatic oxytocin at the sample we acquired at baseline and immediately before the resting-state fMRI scan (~57 min after the end of the last treatment administration). The full timecourses of the changes in plasmatic oxytocin after each method/route of administration can be found elsewhere[11].

**Image acquisition**. MRI data were acquired on a 3 T MR750 system (GE Healthcare, Waukesha, Wisconsin) with a 32-channel head coil. Functional data was acquired using a shot multi-echo echo planar imaging (EPI) sequence (TR = 2500 ms, TE's = 12, 28, 44 and 60 ms, FA = 80°, FOV = 240 × 240 mm², Matrix = 64 × 64, Slice thickness = 3 mm, 32 continuous descending axial slices, resolution = 3.75 × 3.75 × 3 mm). Multi-echo fMRI data acquisition has been shown to optimize sensitivity to detect BOLD signal over noise sources[88]. The resting state fMRI scan lasted 8 min 10 s, with collection of a total of 192 volumes for each echo. A 3D high-spatial-resolution, Magnetisation Prepared Rapid Acquisition (3D MPRAGE) T1-weighted scan was acquired (TR/TE/TI = 7328/3024/400 ms,

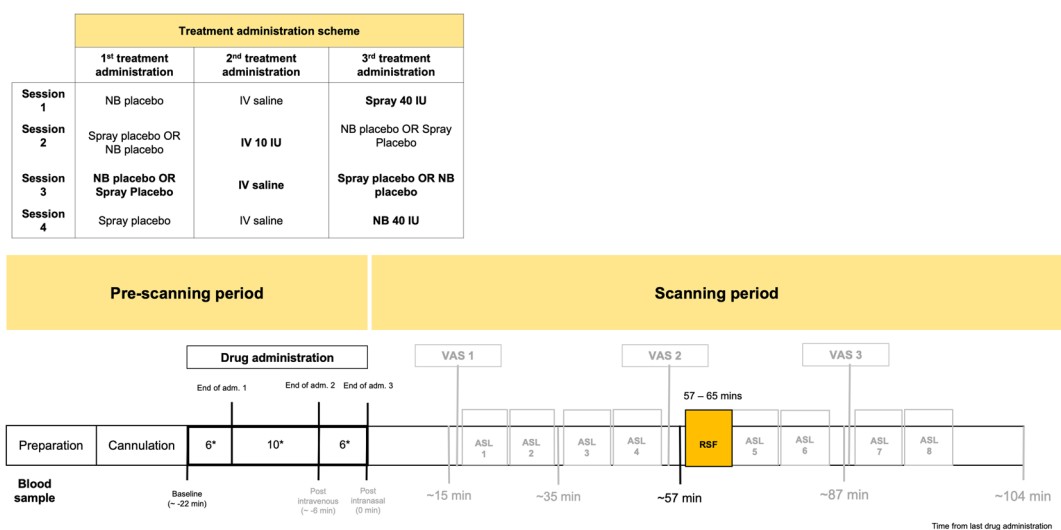

**Fig. 8 Study design, drug administration and blood sampling procedures.** (NB Nebuliser, IV Intravenous, PLC Placebo, pK Pharmacokinetics, RSF Resting-state BOLD-fMRI, VAS Visual Analog Scale). Upper panel—Drug allocation scheme: In each session, participants received treatment via all three administration routes, in one of two fixed sequences: either nebuliser/intravenous infusion/standard nasal spray, or standard nasal spray/intravenous infusion/nebuliser. In three out of four sessions only one route of administration contained the active drug; in the fourth session, all routes delivered placebo or saline. Unbeknown to the participants, the first treatment administration method in each session always contained placebo, while intranasal (spray or nebuliser) oxytocin was only delivered with the third treatment administration. The second administration was an infusion of either saline or oxytocin. Lower panel—Study protocol: After drug administration, participants were guided to the MRI scanner, where eight pulsed continuous arterial spin labelling scans (results reported elsewhere[11]) and one resting-state BOLD-fMRI (RSF) were acquired, spanning 15–104 min after last drug administration offset. We assessed participants' levels of alertness and excitement using visual analogue scales (VAS) at three different timepoints during the scanning session to evaluate subjective drug effects across time (results reported elsewhere[11]). Intranasal administrations lasted for about 6 min, while the intravenous administration occurred at the highest rate of 1IU/min for 10 min. We collected plasma samples at baseline and at five timepoints post-dosing to measure changes in the concentration of oxytocin. The resting-state BOLD-fMRI scan used in this study was acquired between 57–65 min post last drug administration offset (the end of our second intranasal administration was considered time = 0). *Duration of treatment administration in minutes.

FA = 11°, FOV = 270 × 270 mm², slice thickness = 1.2 mm, slice gap = 1.2 mm, matrix = 256 × 256, resolution = 1.1 × 1.1 × 1.2 mm).

**Multi-echo resting-state fMRI data preprocessing and denoising**. Multi-echo fMRI data preprocessing was performed using the AFNI[89] tool meica.py[90,91]. Preprocessing steps included volume realignment, time-series despiking and slice time correction. Functional data were then optimally combined (OC) by taking a weighted summation of the first three echoes, using an exponential T2* weighting approach[92]. While we acquired four echoes, we decided to combine only the first three echoes since the inclusion of the fourth echo can introduce artefacts due to its low signal-to-noise ratio. The Multi-Echo ICA approach included in the meica.py tool was used to remove from the OC data motion artefacts and other sources of noise; this approach has shown greater efficacy in reducing non-BOLD sources of noise and increasing the temporal signal-to-noise ratio when compared to other denoising approaches[64]. White matter (WM) and the cerebrospinal fluid (CSF) mean signals were then extracted from each participant's preprocessed datasets using eroded WM and CSF masks co-registered to each individual's functional space[93]. We then regressed out the average WM and CSF signals from the functional data and applied linear detrending and band-pass filtering (0.01–0.1 Hz) to remove low- and high-frequency noise related to scanner drifts and cardiorespiratory activity respectively[88], and isolate the slow-frequency, spontaneous BOLD fluctuations of interest[94]. We used the six rigid-body parameters extracted for each participant using MCFLIRT[95] to calculate the mean frame-wise displacement (FD). No scrubbing was applied since all participants' motion lied within an acceptable range (<3 mm translation, <3 degrees rotation) for all sessions. We built a study-specific template representing the average T1-weighted anatomical image using the Advanced Normalization Tools (ANTs) toolbox[96]. Finally, each participant's cleaned datasets were co-registered to their corresponding structural scan, normalized to the study-specific template before warping to standard MNI152 space, with 2-mm³ resampling. The final normalised images were visually inspected to ensure the quality of the preprocessing and the absence of artefacts.

**Functional connectivity**. We used the CONN functional connectivity toolbox[97] to generate brain-wide functional connectivity matrices for each subject/session. We extracted the BOLD signal time course for each of the 132 anatomical regions-of-interest (ROIs) included in the default atlas of the CONN toolbox for each participant/session and generated bivariate Pearson's correlation matrices for the

timecourses of all possible pairs of ROIs (132 × 132). The default atlas of the CONN toolbox includes 132 non-overlapping ROIs spanning cortical and subcortical areas and the cerebellum using a combination of the cortical and subcortical areas from the FSL Harvard-Oxford atlas and of the cerebellar areas from the Automatic Anatomical Labelling atlas (Supplementary data 1 for a detailed description). This set of ROIs allows a good whole-brain coverage and has been shown to perform well in comparative brain-wide connectivity analyses[98]. Finally, we normalized our correlation measures to z-scores using the Fisher's r-z transform per subject/session.

**Graph-estimation and network characterization**. We then used our z-transformed connectivity matrices to construct brain graphs representing a functional connectivity network for each subject/session. We used an undirected signed weighted approach in which we kept information about the sign and weight of each of the connections between the nodes (ROIs) of our network, without making assumptions about the direction of this relationship[16]. These connectivity matrices represent fully connected networks, where each node corresponds to each of the ROIs we used to build the functional connectivity matrices and each edge represents the z-transformed value of functional connectivity between a pair of nodes. Recent evidence has demonstrated that retaining the information on the edge weights not only is critical for an accurate understanding of the underlying biology of neural systems[99], but also improves the robustness of graph-related calculations[100]. We used the fully weighted networks to calculate equi-sparse networks by retaining a fixed percentage (K) of the total amount of possible edges. In order to determine systematic pharmacological effects on the network's topological organization that are not dependent on the choice of an arbitrary threshold, we chose a range of sparsity thresholds from 5 to 34%, with steps of 1%, based on previous evidence showing the characteristic small-world behaviour of human brain networks is most consistently observed for this range[30]. For each network and sparsity level, we used the MATLAB functions provided with the Brain Connectivity Toolbox[16] to compute four global and regional network metrics: global efficiency, node degree, local efficiency and betweenness-centrality. Details on the calculation of each of these metrics have been described elsewhere[16]. The combination of these four metrics allows us to test for treatment effects on connectivity at both global and regional levels, taking into account complementary aspects of functional integration of information exchange within the brain network[23]. For a brief overview of the definition and interpretation of each of these graph metrics, please see Supplementary Figure S2.

For each of the graph metrics analysed, we summarized the different values over the range of thresholds using the area under the curve (AUC), instead of applying multiple testing for each metric/threshold. The AUC provides a summary estimator for each network metric and has been shown to be sensitive to topological alterations of brain networks, for instance, as observed with psychiatric disorders[101,102]. All of our statistical analyses on graph metrics were therefore run on AUC parameters rather than the raw values.

**Functional decoding**. We applied reverse inference to gain insight about the potential behavioural relevance of the pathways showing alterations in regional functional connectivity after oxytocin. Binary masks for each of the ROIs where we found significant effects of oxytocin were combined in a single binary map. We then uploaded this combined map onto NeuroSynth[103] to match the co-activation pattern with the other studies in the database using the decoder function (http://neurosynth.org/decode/). We downloaded the top 100 terms ranked by correlation strength between the combined map and the meta-analytic one from decoder. As per a previous study[104], the terms related to brain regions or imaging methods were excluded. We present these decoding results as a word cloud, where font size codes correlation strengths.

**Overlap with large-scale resting-state networks**. To improve the interpretability of our findings and facilitate comparisons with previous work focusing on large-scale resting-state networks[19,20], we calculated the percentage of overlap between our single binary map including all regions showing alterations in regional functional connectivity after oxytocin (irrespective of route/method of administration) and the large-scale resting-state networks described in the atlas from Yeo et al (2011)[105]. This atlas includes a coarse parcellation of seven canonical resting-state networks, namely the default-mode, dorsal attention, frontoparietal, limbic, sensorimotor, visual and ventral attention networks. Overlap was quantified using the Dice coefficient, which estimates the percentage of voxels of each resting-state network that overlap with our treatment effect mask.

**Statistics and reproducibility**

*Effects of exogenous oxytocin on head movement*. Mean FD was compared across the four treatment groups to confirm that there were no group differences in head motion, using repeated measures one-way analysis of variance implemented in SPSS 24 (http://www-01.ibm.com/software/uk/analytics/spss/), with the Greenhouse-Geisser correction against violations of sphericity (Supplementary Fig. S3).

*Effects of exogenous oxytocin on the functional connectome*. We investigated the effects of oxytocin on the functional connectome at rest considering different scales of its organization: macro-, meso- and microscales as suggested by Brown et al.[106]. We did so by examining treatment-related changes in mean connectivity and global efficiency of the whole-brain network (macroscale), graph properties of regional connectivity at the node level (mesoscale) and individual connections between pairs of nodes (microscale). These analyses were performed using repeated measures one-way analysis of variance, implemented in Matlab 2016. When a significant effect was found, we followed up with post-hoc paired t-tests between each pair of the four treatment conditions, using FDR correction for multiple testing.

Macroscale. Mean functional connectivity was calculated as the average of the elements in the lower triangular connectivity matrix for each participant and treatment condition. Additionally, we retrieved the global efficiency of the brain network for each participant and treatment condition. In both cases, we compared treatment groups using repeated measures one-way analysis of variance.

Mesoscale. We retrieved values for the degree, betweenness-centrality and local efficiency of each node of the connectome for each subject and treatment condition. We first compared the four treatment conditions using repeated measures one-way analysis of variance, controlling false positives with FDR correction for the number of nodes examined.

Microscale. We performed two different analyses to investigate treatment-related changes on each individual edge of the connectivity matrices. First, we performed comparisons between the four treatment conditions on each element of the lower triangle of the connectivity matrices, controlling false positives with an FDR correction for the number of connections. Second, since this kind of correction is highly conservative, for completeness, we also run the same analysis but used the Network Based Statistics (NBS) method implemented in the "Network Based Statistic Toolbox v1.2" (NBS toolbox) (The University of Melbourne, Melbourne, Australia)[107] to control the family-wise error rate (in the weak sense) when mass-univariate testing is performed at every edge of the graph. While this approach does not allow for inferences on individual connections, it allows the extraction of subnetworks or topological clusters of regions that are significantly differently connected between conditions[107]. This method has been specifically designed to perform statistical analysis at the connectome level. When compared to analyses at the individual edge level, the NBS method offers greater sensitivity, while also controlling for false positive discovery[107]. In our NBS analysis, we tested a range of arbitrary primary thresholds varying from 1.5 to 4, with increments of

0.5. We then used a two-tailed secondary threshold of 0.025 (0.05/2), with 5000 permutations.

*Plasmatic concentrations of oxytocin*. Finally, we wanted to understand whether the plasmatic oxytocin concentrations at the time the resting-state scan was acquired (57 min after the end of the last treatment administration) differed across the four treatment conditions. For this purpose, we conducted a linear mixed model (implemented in SPSS) on plasmatic oxytocin concentrations measured in a blood sample collected immediately before the beginning of the resting-state acquisition, using Time (baseline, oxytocin concentration at 57 min post-administration), Treatment and Time × Treatment as fixed factors and a random intercept for subject. We followed up the significant interaction with further post-hoc tests to examine differences between each pair of treatment conditions between baseline and post-administration and between treatment conditions within each time point, using Tukey's correction for multiple testing. A full report on the pharmacokinetics of exogenous oxytocin during the full scanning protocol for each method of administration has been published elsewhere[11].

For all of our analyses, we set statistical significance to $p < 0.05$, after correction for multiple testing, when applicable. We used the ggplot package from R (version 3.5.3) to build the violin plots and the Matlab toolbox "BrainNet Viewer"[108] to create the figures depicting the effects of treatment condition on the nodal metrics projected in the brain space.

**Reporting summary**. Further information on research design is available in the Nature Research Reporting Summary linked to this article.

## Data availability
Data can be accessed from the corresponding author upon reasonable request. A reporting summary for this Article is available as a Supplementary Information file. The source data underlying Figs. 1 and 7, and Supplementary Figures 1 and 3 are provided as a Source data file (Supplementary Data 2).

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

## Acknowledgements

We would like to thank all participants that volunteered for this study. We would also like to thank Sofia Vasilakopoulou, Jack Loveridge and Ndaba Mazibuko for their assistance in data collection and Dr Fernando Zelaya for his comments on the first draft of the manuscript. This study was part-funded by: an Economic and Social Research Council Grant (ES/K009400/1) to Y.P.; scanning time support by the National Institute for Health Research (NIHR) Biomedical Research Centre at South London and Maudsley NHS Foundation Trust and King's College London to Y.P.; an unrestricted research grant by PARI GmbH to Y.P. O.D. is supported by the National Institute for Health Research (NIHR) Biomedical Research Centre at South London and Maudsley NHS Foundation Trust and King's College London.

## Author contributions

Y.P. designed the study; Y.P. collected the data; D.M. and O.D. analyzed the data; D.M. and Y.P. wrote the first draft of the paper and all co-authors provided critical revisions.

## Competing interests

The authors declare no competing interests. This manuscript represents independent research. The views expressed are those of the authors and not necessarily those of the NHS, the NIHR, the Department of Health and Social Care, or PARI GmbH.
