## [Peer Review File · Communications Biology]

Reviewers' comments:

Reviewer #1 (Remarks to the Author):

The present placebo-controlled pharmacological fmri study examined the effects of three different application routes of oxytocin (intranasal spray, intranasal nebulizer, intravenous) versus placebo on the intrinsic functional connectome as measured by resting state fMRI. The manuscript is well written, and the authors present a clear rationale and balanced discussion of the findings. The study has several major strengths, including the use of three administration routes, an excellent placebo-controlled cross-over design and a state of the art network analytic approach. Nevertheless, I have major concerns with respect to the robustness of the findings and some additional comments as outlined below:

- My major concern refers to the comparably low sample size of $n = 16$. Systematic examination of the effects of sample size on graph metrics in resting state fMRI have been conducted and suggest that sample sizes of around 40 (combined with long scan acquisition) or even 100 subjects (with shorter scan durations) are needed to robustly determine reliable graph theoretical metrics (see e.g. Termenon et al., 2016; 10.1016/j.neuroimage.2016.05.062). Can the authors exclude that the somewhat unsystematic and widespread differences between the treatment conditions are influenced by the lack of robust determination of network-based metrics within each treatment session? The estimation of robust network metrics within each of the treatment conditions (with $n = 16$ individuals, after exclusion of one subject) is critical to interpret the subsequent comparison between the treatment conditions.
- The interpretation of the results with respect to the behavioral effects of oxytocin (line 230-252, in the discussion) is speculative. The authors employ functional decoding via neurosynth and integrating these findings into the discussion would be more appropriate to infer the potential functional relevance of the results.
- The results are rather distributed, and authors interpret the findings in the context of circuits and interacting brain systems. Mapping the identified regions on large scale atlases such as the Yeo atlas could provide a better integration of the findings and allow a more integrative view on the larger scale networks affected by oxytocin in general or the specific administration routes in particular.
- Given the journal style (methods the end) the authors should include design specifics (e.g. within subject design, number of subjects, male sample) in the introduction.
- Participants with 'below 5 cigarettes per day' were included. Nicotine has strong and robust acute and subacute effects on intrinsic brain networks. Did the authors control for differences in smoking between the sessions?
- In the context of the extensive literature reporting effects of intranasal oxytocin on brain functional connectivity during rest and the own recent studies of the authors the lack of clear hypotheses in the introduction is surprising. Inclusion of clear hypotheses in the introduction would strengthen the interpretation of findings.

Minor

- Sample size and sex of the participants should be clearly indicated in the abstract, title should reflect that only males were included

Reviewer #2 (Remarks to the Author):

Review for Oxytocin modulates local topography of human functional connectome at rest but not global capacity for information transfer

Overview

I congratulate the authors on writing a very interesting paper that I enjoyed reading. Despite concerns about sample size and different drug doses achieving different oxytocin concentrations, I believe the paper makes an interesting and necessary contribution to the literature. My comments focus on the transparency of the entire study design and procedure, as well as specific details regarding participants and drug administration.

Comments:

Abstract

1. Is concise and well-written. However, in order to be more transparent about the sample size used, it would be useful to add the sample size here for readers to see up front.

Introduction

2. Is well-written and provides a clear description of why the research is of interest and why the methods used were chosen.

Results

3. Are clear and transparent. Although see my suggestion in the Methods section about including an oxytocin AUC analysis as well as an individual sample analysis at 57 minutes.

Discussion

4. Provides an accurate summary of the key findings along with an interesting discussion and suggestions for future research, while acknowledging the study limitations.

Materials and Methods

5. The authors list a number of interesting inclusion/exclusion criteria, without full explanation for why they thought this was necessary. For example, participants were required to provide a urine test for recreational drug use and to avoid exercise. Were these chosen in relation to MRI or OT? Some clarity would be useful in order for others to judge whether this is something that should be widely adopted. The authors also only recruited males to the study – as is typical – however, they fail to explain why this decision was made; presumably due to ethical concerns regarding intravenous OT to women, however, this should be explicitly stated. (I note that the authors highlight this as a limitation of their study.)

6. Although the authors describe the study design as double-blind, it seems reasonable to assume that the experimenter would have known that the first spray was a placebo. Can the authors confirm if that assumption is wrong and why? Otherwise it may be more accurate to describe the study as partially double-blind.

7. A clear explanation is given for why 40IU was chosen as a dosage, despite this not being the typical dosage in behavioural studies – although the useful description that this high dose was chosen to avoid attributing null effects to under dosing is also worth including here as well as in the discussion. However, more detail should be given about the administration itself. Were participants supervised during intranasal drug administration? Were the guidelines outlined by Guastella and colleagues (2013) followed or alternative in-house guidelines?

8. If the authors were aware in advance that the intravenous dosage of oxytocin used was likely to generate significantly higher oxytocin concentrations, some justification for this is required in the methods section as well as a discussion in the limitations. Are they in the best position to compare 'like for like' when achieving different dosages?

9. It is not clear from the text when drug administration was relative to blood sampling. Please clarify the exact timeline of the sessions.

10. Figure 1 needs more clarity. The numbers underneath presumably represent when blood samples were taken (this could be made clearer), but it would also be helpful to see explicitly at what time after drug administration these were (on average). For example, sample 3 was taken, immediately after drug administration (or 6 mins depending on the session?), but when was sample 4 taken etc? Was sample 1 the baseline for every participant regardless of session? From the text one is able to infer that 6, 10 & 6 refer to the duration of drug administration, but this could also be made more explicit. VAS presumably stands for Visual Analogue Scale, but this is not referenced in the figure caption.

11. I haven't personally come across the idea of a word cloud to present the potential behavioural relevance of oxytocin's effects on connectivity, but I liked the idea and found it very interesting.

12. It seems strange to anticipate different head movements based on drug condition, has this been found previously? If not, perhaps move the whole analysis to supplementary materials and briefly mention in the data pre-processing subsection.

13. In the result section the authors are explicit about which blood sample was used to compare effects of administration techniques on oxytocin concentrations, but it should also be stated explicitly in the methods section as well. Given that existing literature suggests there are individual differences in baseline oxytocin concentrations, did the authors consider using the baseline sample in their analyses? It would also be interesting to know whether AUC oxytocin up to sample 6 differed across treatment groups (from Fig 6.a of the authors other 2020 paper this doesn't seem likely, but as this analysis was run on the whole session not up to sample 6, it is not possible to say for certain).

Minor comments:

Pg. 3, line 69 – Should read "Although these types of studies are..."

Pg. 10, line 336 – Should read "... these differences between the..."

Pg. 12 – In the PARI SINUS nebuliser subsection, missing a % mark after 15.7

Pg. 15 - missing a % mark after 5

Pg. 16 – In the Mesoscale subsection, don't need to repeat the final sentence, as this is already stated in the overview.

Supplementary materials, Table S1 – delete 'ed' from regions-of-interested.

**Department of
Neuroimaging**

Reviewers' comments:

Reviewer #1 (Remarks to the Author):

The present placebo-controlled pharmacological fmri study examined the effects of three different application routes of oxytocin (intranasal spray, intranasal nebulizer, intravenous) versus placebo on the intrinsic functional connectome as measured by resting state fMRI. The manuscript is well written, and the authors present a clear rationale and balanced discussion of the findings. The study has several major strengths, including the use of three administration routes, an excellent placebo-controlled cross-over design and a state of the art network analytic approach. Nevertheless, I have major concerns with respect to the robustness of the findings and some additional comments as outlined below:

1. My major concern refers to the comparably low sample size of $n = 16$. Systematic examination of the effects of sample size on graph metrics in resting state fMRI have been conducted and suggest that sample sizes of around 40 (combined with long scan acquisition) or even 100 subjects (with shorter scan durations) are needed to robustly determine reliable graph theoretical metrics (see e.g. Termenon et al., 2016; 10.1016/j.neuroimage.2016.05.062). Can the authors exclude that the somewhat unsystematic and widespread differences between the treatment conditions are influenced by the lack of robust determination of network-based metrics within each treatment session? The estimation of robust network metrics within each of the treatment conditions (with $n = 16$ individuals, after exclusion of one subject) is critical to interpret the subsequent comparison between the treatment conditions.

The reviewer asks if this sample size would be sufficient to estimate graph theory metrics of functional connectivity with sufficient robustness. Indeed, the impact of several methodological choices, including those related to sample size, on the reliability of graph-theory metrics to characterize functional connectivity at rest has been highly debated (for some reviews on this topic please see ^{1,2}). While some studies provided evidence that these metrics can be reliably measured across a wide range of sample sizes ($5 < n < 45$)², the reviewer points out a study that raises concerns about the use of sample sizes smaller than 100 participants (or 40 participants, if the acquisitions are longer). The study pointed out by the reviewer makes an extremely important contribution to the field by providing estimates of minimal sample sizes and durations of acquisition that maximize reliability when functional connectivity is studied using graph theory metrics³. We believe though that the direct extrapolation of their estimates to our study is not necessarily warranted for two main reasons. First, *Termenon et al. (2016)* relied on *single* echo data pre-processed with a suboptimal denoising pipeline (namely, the fMRI signal in each voxel was simply weighted by the probability of each voxel belonging to grey matter; the authors decided to not regress out signal from white matter (WM) or cerebrospinal fluid (CSF)). However, the reliability of network-based metrics has been shown to be negatively affected by inefficient noise removal from the BOLD signal^{1,2}. Second, the authors relied on unweighted binary graphs, which do not take into account the sign and magnitude of the relationship between each pair of nodes. The use of unweighted graphs is associated with lower reliabilities of graph theory metrics of connectivity¹. Hence, it is not clear how the suggestion in *Termenon et al. (2016)* generalises to datasets where acquisition parameters and enhanced data pre-processing improve data quality and signal to noise ratio, and more stringent analytic practices are adopted. Please see also *Kolosa et Kopp (2018)* for recent discussions on the importance of data quality over data quantity⁴.

Department of Neuroimaging

In our study, we decided to use multi-echo data, which we denoised by using a data-driven approach (ME-ICA) and by additionally regressing out the signals from the WM and CSF. The combination of multi-echo data with this denoising pipeline has been shown to provide better SNR as compared to single-echo acquisitions or multi-echo acquisitions denoised with other suboptimal approaches⁵, i.e. a standard regression of motion parameters and WM and CSF signals. Moreover, we relied on weighted graphs to further maximize reliability. Furthermore, we also estimated the graph metrics using the areas under the curve of each metric across different costs, which provides a more robust estimate of each metric independently of specific choices for the cost used to build the graphs. Indeed, *Termenon et al. (2016)* also demonstrate how the reliability of these metrics is heavily influenced by cost. We agree that further work needs to be done to investigate the impact of methodological choices that improve data quality on the reliability of network-based metrics. In the absence of a systematic investigation of such analytic choices, we would argue that the enhanced data quality (in terms of SNR and selection of network metrics) that we employed in our study would only increase the robustness of our findings compared to the minimum requirements described in *Termenon et al. (2016)* for datasets that followed less robust data-cleaning pipelines and graph metric selection processes. Mindful of this limitation, we call the attention of the reader to this fact in our Discussion of the limitations of the study (see below). Furthermore, we openly label the study as exploratory in many instances of the manuscript.

Line 363: “Our study faces certain limitations that we should acknowledge. Firstly, while our study provides proof-of-concept of the potential of connectomics as a new approach to study the pharmacodynamics of exogenous oxytocin in the human brain, our sample size was not optimized to capture effects of oxytocin on the connectome smaller than a medium effect size. Hence, it is possible we might have missed smaller effects for at least some of the different dependent variables examined here (this limitation may have particularly affected our analysis on individual edges, where we applied stringent control for multiple-testing across the number of edges). There has been discussion about the minimal sample size and amount of data required to estimate graph metrics of functional connectivity with sufficient robustness^{1,2}. While some studies provided evidence that these metrics can be reliably measured across a wide range of sample sizes ($5 < n < 45$)², another study has suggested that only sample sizes in the order of 100 individuals (40 individuals if combined with long fMRI acquisitions) are enough to estimate graph metrics with sufficient reliability³. However, this study used a suboptimal denoising pipeline which could have hindered the reliability of the estimated graph metrics. Indeed, studies have shown that different methodological options related to, for instance, the type of cleaning of BOLD signal used and the type of connectivity matrices used to estimate the graphs (weighted versus unweighted) can affect the reliability of these metrics. In this study, we attempted to maximize the reliability of the graph metrics we used by acquiring multi-echo data and applying a denoising pipeline that has been shown to provide better signal-to-noise ratio as compared to single-echo acquisitions or multi-echo acquisitions denoised with other suboptimal approaches⁵, i.e. a standard regression of motion parameters and WM and CSF signals. Moreover, we relied on weighted graphs to further maximize reliability. Furthermore, we also estimated the graph metrics using the areas under the curve of each metric across different costs, which provides a more robust estimate of each metric independently of specific choices for the cost used to build the graphs. Further work needs to be done to investigate the impact of methodological choices that improve data quality on the reliability of network-based metrics, such as those we used here. Irrespectively of this, future studies should attempt to replicate our approach with larger samples, preferentially applying a dose-response design and sampling different intervals post-dosing.”

2. The interpretation of the results with respect to the behavioral effects of oxytocin (line 230-252, in the discussion) is speculative. The authors employ functional decoding via neurosynth and integrating these findings into the discussion would be more appropriate to infer the potential functional relevance of the results.

Thanks for the suggestion. Indeed, we did not make full justice to our functional decoding analysis. We have now revised the discussion to incorporate these findings.

Line 252: “Since our data was acquired at rest, inferences on the behavioural relevance of the observed effects should be made with caution. However, it seems that, overall, oxytocin can prime, at rest, neural circuits that have been described to contribute to a number of mental/behavioural processes – that we summarized through functional decoding using the Neurosynth database.”

Department of Neuroimaging

These processes were mostly related to early visual processing, mentalizing and theory-of-mind, social and audiovisual processing, facial processing, expectancy, aversion, somatosensation (pressure and nociception) and heart physiology. Putting all of these processes together, we suggest that an overarching model for the effects of oxytocin on human behaviour could be summarized in the following three main mechanisms: 1) oxytocin may facilitate the processing of socially relevant sensory cues and enhance our capacity to create cognitive models of others' mental and affective states, which together may concur to adjust behaviour according to the dynamic interplay between sender (others) and receiver (self)⁶; 2) oxytocin may facilitate approach behaviour towards others, by enhancing the motivational value of social interactions and decreasing approach avoidance/anxiety⁷; 3) oxytocin might facilitate the integration of cognitive processing with physiological signals, contribute to the encoding of the saliency or precision of interoceptive signals, and facilitate the acquisition of generative brain models of the emotional and social selves⁸. “

3. The results are rather distributed, and authors interpret the findings in the context of circuits and interacting brain systems. Mapping the identified regions on large scale atlases such as the Yeo atlas could provide a better integration of the findings and allow a more integrative view on the larger scale networks affected by oxytocin in general or the specific administration routes in particular.

Thanks for the suggestion. We agree this further analysis would bring some further insight to the interpretation of our findings and help direct comparisons with previous hypothesis-driven studies on large-scale networks^{9,10}. Hence, we used the brain mask containing all regions-of-interest where we found treatment effects (irrespective of route/method of administration) which we used for our functional decoding analysis to calculate the percentage of overlap of this mask with each of the seven networks in the Yeo atlas¹¹. The extent of overlap of our mask with each of the seven networks was quantified using the Dice coefficient. Please see below all changes we implemented in the revised manuscript as a result of this further analysis.

Methods section

Line 577: “Overlap with large-scale resting-state networks

To improve the interpretability of our findings and facilitate comparisons with previous work focusing on large-scale resting-state networks^{9,10}, we calculated the percentage of overlap between our single binary map including all regions showing alterations in regional functional connectivity after oxytocin (irrespective of route/method of administration) and the large-scale resting-state networks described in the atlas from Yeo et al (2011)¹¹. This atlas includes a coarse parcellation of seven canonical resting-state networks, namely the default-mode, dorsal attention, frontoparietal, limbic, sensorimotor, visual and ventral attention networks. Overlap was quantified using the Dice coefficient, which estimates the percentage of voxels of each resting-state network that overlap with our treatment effect mask .”

Results section

Line 202: “Overlap with large-scale resting-state networks

The pathways for which we identified significant modulatory effects of oxytocin overlapped primarily with regions belonging to the visual network (Dice coefficient=0.68), followed by the frontoparietal (Dice coefficient=0.54), default-mode (Dice coefficient=0.49), limbic (Dice coefficient=0.48) and ventral attention networks (Dice coefficient=0.44). The overlap with the dorsal attention (Dice coefficient=0.20) and sensorimotor (Dice coefficient=0.24) networks was considerably lower (Figure 6).”

Figures

**Department of
Neuroimaging**

“Fig. 6 – Overlap between the neural circuits engaged by exogenous oxytocin at rest and the canonical large-scale resting-state networks. We calculated the percentage of overlap between our single binary map including all regions showing alterations in regional functional connectivity after oxytocin (irrespective of route/method of administration) and the large-scale resting-state networks described in the atlas from Yeo et al. (2011)¹¹. Overlap was quantified using the Dice coefficient, which measures the percentage of voxels of each resting-state network overlapping with our treatment effect mask. In the upper panel A, we provide an overview of all regions where we found treatment effects, irrespective of route/method of administration, rendered in a 3D surface model. In the lower panel B, we provide a heatmap summarizing the percentage of overlap (Dice coefficient) between the regions in A and each of the seven networks (each network rendered in a 3D surface model). DMN-Default-mode network; DA – Dorsal attention; FPN – Frontoparietal network; SMN – Sensorimotor network; VA – Ventral attention.”

Discussion section

Line 240: “At first glance, the regional effects we found seem dispersed across many brain regions, without a very clear organizational pattern. Nevertheless, a careful examination of these changes suggests that some of these regions are part of well-described neuronal circuits that operate together in a number of brain functions. Indeed, the brain pathways we found to be modulated by exogenous oxytocin overlapped considerably with a number of well described canonical large-scale resting-state networks, namely the visual, frontoparietal, default-mode, limbic and ventral attention networks. We note that two previous studies applying hypothesis-driven analyses have reported consistent effects of a single dose of intranasal oxytocin on the default-mode, salience and attention networks, including on their between-network connectivity (other networks were simply not investigated)^{9,10}. Hence, while our data is broadly compatible with these two studies, we expand their conclusions in, at least, one important direction. We show that most likely the effects of oxytocin on brain processing involve several brain networks and can hardly be attributed to a single neural pathway. Instead, oxytocin most likely operates at multiple levels, spanning basic sensory processing to salience detection and attention orientation, emotional processing and high-order mental representations of the self and others.”

4. Given the journal style (methods the end) the authors should include design specifics (e.g. within subject design, number of subjects, male sample) in the introduction.

Thanks for the suggestion. This has been added to the Introduction. The revised text reads as below.

Line 116: “In this exploratory study, we administered 40IU intranasally (using either a standard nasal spray or a nebuliser) or 10 IU intravenously to 16 healthy men using a placebo-controlled, double-blind, triple-dummy, crossover design. We acquired multi-echo resting-state BOLD fMRI data and used graph-theory modelling, adopting a data-driven approach, to explore the changes in the global and regional topography of the human functional connectome that follow the administration of exogenous oxytocin through both intranasal and intravenous routes, compared to placebo.”

5. Participants with ‘below 5 cigarettes per day’ were included. Nicotine has strong and robust acute

**Department of
Neuroimaging**

and subacute effects on intrinsic brain networks. Did the authors control for differences in smoking between the sessions?

As the reviewer points out, we excluded participants with heavy smoking habits to minimize the overall impact that nicotine or other cigarette components could have on the physiology of the brain. As a further clarification, only 3 participants were regular smokers (2-3 cigarettes/day) and 1 participant used to smoke occasionally (around 5 cigarettes/month) at the time of the study. In any case, to control for differences in smoking between sessions, we further asked participants to restrain from smoking or drinking any beverage other than water in the two hours preceding the session. These aspects have now been clarified in the revised version of the manuscript.

Line 426: "Participants were required to consume <28 units of alcohol per week and <5 cigarettes per day. We further instructed participants to abstain from alcohol and heavy exercise for 24 hours before scanning, and to restrain from any beverage or food consumption or smoking (if they were smokers) for 2 hours before scanning. Intense physical exercise¹² and food intake¹³ have been shown to impact on oxytocin release and could have introduced additional noise. Furthermore, nicotine has robust acute¹⁴ and subacute¹⁵ effects on brain's physiology, which could have affected our neuroimaging metrics in unpredictable ways. Only three participants were regular smokers (2-3 cigarettes/day) and one participant used to smoke occasionally (around 5 cigarettes/month) at the time of the study."

6. In the context of the extensive literature reporting effects of intranasal oxytocin on brain functional connectivity during rest and the own recent studies of the authors the lack of clear hypotheses in the introduction is surprising. Inclusion of clear hypotheses in the introduction would strengthen the interpretation of findings.

The reviewer is right when he/she says that we could have generated *a priori* hypothesis based on previous studies on the effects of intranasal oxytocin on functional connectivity at rest using standard nasal sprays. However, testing specific hypotheses was not from the beginning the main purpose of this study for two reasons.

First, our study is the first of its kind to investigate the effects of intravenous or intranasal administration using a nebuliser on functional connectivity at rest. Even our previous findings on the effects of different routes of administration on brain perfusion at rest could hardly inform the hypotheses for this study, since the previous study had focused on oxytocin-induced effects on local perfusion in the brain. Effects on local perfusion do not take into account interactions between brain regions, which we effectively investigated in the connectomic analyses presented in the current manuscript (please note we have made now more explicit the novelty of this work in relation to our previous perfusion work – see below).

Second, by looking at the effects of oxytocin on functional connectivity at rest, in this study we intended to go beyond the brain regions typically investigated in previous studies, which have reasonably mostly focused on the amygdala as a key-hub of the oxytocin system in the brain but have not considerably contributed to expand our view of the effects of oxytocin on the brain. Hence, here we primarily intended to generate new hypotheses regarding the brain substrates targeted by oxytocin when administered using different routes. Looking at our findings, our data-driven exploratory approach seems to have been advantageous. Indeed, we found effects on areas already described in previous studies, such as the amygdala. However, in addition we also unravelled a number of other effects on other brain regions, which are novel and will inform future studies. We hope the reviewer agrees that since testing specific hypotheses was not a primary goal defined for this study *a priori*, defining now hypotheses *post hoc* after we have seen the results of the exploratory analyses would not be appropriate since we would be inevitably biased. In any case, we now describe the goals of our study more clearly in the revised version of the Introduction.

Line 136: "In our last perfusion study, we investigated oxytocin induced effects on local perfusion in the brain¹⁶, which do not take into account interactions between brain regions, a core feature of the biology of the brain¹⁷. Hence, in this study we sought

**Department of
Neuroimaging**

to expand our previous perfusion findings and apply connectomic analyses to investigate how exogenous oxytocin, when administered using different routes and methods for intranasal administration, impacts on the functional connectome. Rather than testing specific hypotheses, we conducted exploratory connectomic analyses at the whole-brain, as an attempt to unravel new aspects of the effects of oxytocin on the brain that previous studies might have missed by simply focusing on key-hubs of the brain oxytocin system, such as the amygdala."

Minor

7. Sample size and sex of the participants should be clearly indicated in the abstract, title should reflect that only males were included

Thanks for the suggestion. We have implemented the suggested changes in revised version of the title and abstract.

Reviewer #2 (Remarks to the Author):

Review for Oxytocin modulates local topography of human functional connectome at rest but not global capacity for information transfer

Overview

I congratulate the authors on writing a very interesting paper that I enjoyed reading. Despite concerns about sample size and different drug doses achieving different oxytocin concentrations, I believe the paper makes an interesting and necessary contribution to the literature. My comments focus on the transparency of the entire study design and procedure, as well as specific details regarding participants and drug administration.

Thanks very much for the positive appraisal of our manuscript.

Comments:

Abstract

1. Is concise and well-written. However, in order to be more transparent about the sample size used, it would be useful to add the sample size here for readers to see up front.

The sample size has now been added to the abstract.

Introduction

2. Is well-written and provides a clear description of why the research is of interest and why the methods used were chosen.

Results

3. Are clear and transparent. Although see my suggestion in the Methods section about including an oxytocin AUC analysis as well as an individual sample analysis at 57 minutes.

Discussion

4. Provides an accurate summary of the key findings along with an interesting discussion and suggestions for future research, while acknowledging the study limitations.

Materials

and

Methods

5. The authors list a number of interesting inclusion/exclusion criteria, without full explanation

**Department of
Neuroimaging**

for why they thought this was necessary. For example, participants were required to provide a urine test for recreational drug use and to avoid exercise. Were these chosen in relation to MRI or OT? Some clarity would be useful in order for others to judge whether this is something that should be widely adopted. The authors also only recruited males to the study – as is typical – however, they fail to explain why this decision was made; presumably due to ethical concerns regarding intravenous OT to women, however, this should be explicitly stated. (I note that the authors highlight this as a limitation of their study.)

Thanks for the opportunity to provide this further information. Indeed, we had not explained our inclusion/exclusion criteria with sufficient detail. We have now revised the text to explicitly state the rationale for our decisions regarding inclusion/exclusion of participants.

Line 419: *We enrolled 17 healthy male adult volunteers (mean age 24.5, SD = 5, range 19-34 years). One participant did not complete one of the four visits and for this reason was excluded from all analyses. We decided to focus our study on men because the levels of oxytocin have been shown to fluctuate across the menstrual cycle in women, which would introduce an additional level of unwanted variability in our sample¹⁸. We screened participants for psychiatric conditions using the Symptom Checklist-90-Revised¹⁹ and the Beck Depression Inventory-II²⁰ questionnaires. Participants were not taking any prescribed drugs, did not have a history of drug abuse and tested negative on a urine panel screening test for recreational drugs. Recreational drugs, such as MDMA, have been shown to increase the release of oxytocin to the plasma^{21,22}, which could have impacted on our findings. Participants were required to consume <28 units of alcohol per week and <5 cigarettes per day. We further instructed participants to abstain from alcohol and heavy exercise for 24 hours before scanning, and to refrain from any beverage or food consumption or smoking (if they were smokers) for 2 hours before scanning. Intense physical exercise¹² and food intake¹³ have been shown to impact on oxytocin release and could have introduced additional noise. Furthermore, nicotine has robust acute¹⁴ and subacute¹⁵ effects on brain physiology, which could have affected our neuroimaging metrics in unpredictable ways. Only three participants were regular smokers (2-3 cigarettes/day) and one participant used to smoke occasionally (around 5 cigarettes/month) at the time of the study.”*

Line 451: *“All sessions were conducted in the morning to minimise potential circadian variability in resting brain activity²³ or oxytocin levels²⁴.”*

6. Although the authors describe the study design as double-blind, it seems reasonable to assume that the experimenter would have known that the first spray was a placebo. Can the authors confirm if that assumption is wrong and why? Otherwise it may be more accurate to describe the study as partially double-blind.

The reviewer is correct in pointing out that the experimenter was aware that the first of the three administrations in each of the four visits was always a placebo (while the participants were not aware of this fact). Nonetheless, we believe that the study remains a double-blind study as the experimenter was blind regarding whether, in any given visit, the participant had received active spray oxytocin, active nebuliser oxytocin, active intravenous oxytocin, or placebo.

We used this triple dummy procedure where participants, in each visit, experienced all three administration methods, sequentially, only one of which was active (and with the intravenous administration always being in the middle). By keeping the first administration (intranasal, spray or nebuliser) always placebo, we achieved two goals: first, we avoided the possibility of washing out the active intranasal treatment if it was administered first, followed by a nasal placebo treatment as the third administration; second, we maintained post-intranasal dosing timings consistent. The experimenter was blind as to the treatment allocation plan for a given session (i.e. whether administration 3 was active intranasal oxytocin or

**Department of
Neuroimaging**

placebo). Additionally, since no measures were acquired between the first administration (spray or nebuliser, always placebo) and the second administration (intravenous), we do not believe that the fact that the experimenter was aware that the first session was placebo could have introduced any bias.

7. A clear explanation is given for why 40IU was chosen as a dosage, despite this not being the typical dosage in behavioural studies – although the useful description that this high dose was chosen to avoid attributing null effects to under dosing is also worth including here as well as in the discussion. However, more detail should be given about the administration itself. Were participants supervised during intranasal drug administration? Were the guidelines outlined by Guastella and colleagues (2013) followed or alternative in-house guidelines?

Regarding the intranasal oxytocin dose, it is worth noting that the 24 IU is indeed popular in oxytocin studies, but not exclusive, and generally the selection of dose in oxytocin studies has not been informed by detailed dose-response characterizations. Our choice of 40IU was made for the purposes of matching our previous work on the pharmacodynamics of oxytocin in healthy volunteers²⁵, and is a dose we²⁶⁻³⁴ and others (e.g. ³⁵) have commonly used with patients. As the reviewer points out, using a high dose would indeed avoid attributing null effects to under dosing – we now allude to this fact explicitly in the revised version of the manuscript (as can be seen below). The intranasal administrations were indeed supervised by the lead experimenter of the study. For the standard nasal spray, we followed our standard in-house protocol that we have used consistently across all of our studies and which is consistent with the guidelines outlined by *Guastella et al. (2013)*³⁶. These two aspects have now been added to the revised version of the manuscript as outlined below.

Line 460: *“Participants self-administered intranasal oxytocin or placebo with the nasal spray or nebuliser following under the supervision of the lead experimenter who ensured the correct application of the devices. Additionally, all participants had been trained during the screening visit in the correct administration procedures.”*

Line 462: *“Nasal spray administration. Participants followed our standard in-house protocol that we have used consistently across all of our studies and is consistent with the guidelines outlined by Guastella et al. (2013)³⁶. Specifically, participants self-administered 10 puffs, each containing 0.1ml Syntocinon (4IU) or placebo, one puff every 30s, alternating between nostrils (hence 40IU OT in total). Participants blocked opposite nostril while administering each puff, and each puff was followed by an immediate, brief, sharp, snort.”*

8. If the authors were aware in advance that the intravenous dosage of oxytocin used was likely to generate significantly higher oxytocin concentrations, some justification for this is required in the methods section as well as a discussion in the limitations. Are they in the best position to compare ‘like for like’ when achieving different dosages?

Thanks for pointing out the lack of sufficient detail in explaining the rationale for our intravenous dose. The reviewer is right that 10 IU (over 10min) in our case increased the concentrations of plasmatic oxytocin beyond those observed for the spray or nebuliser (we reported the full time-course of variations in plasmatic oxytocin in another manuscript we published earlier this year)¹⁶. This was an intentional aspect of our study design. We decided to use the highest intravenous dose (at the highest rate of 1IU/min) that we could get permission to administer safely in healthy volunteers as a proof of concept, so as to achieve a robust and prolonged increase in plasmatic oxytocin over the course of our full testing session. In this manner, we demonstrate that even when plasmatic levels of oxytocin are maintained substantially increased throughout the observation interval, intranasal administrations of oxytocin induce changes in the connectome that the intravenous route cannot produce. Therefore, the reviewer is correct that our design was suboptimal to head-

**Department of
Neuroimaging**

to-head comparisons with the intranasal route. However, this aspect of our design strengthens our confidence in the ability of the intranasal route to target specific regions, since from our data it is clear that the lack of effects of intravenous oxytocin on regions specifically targeted by intranasal methods is unlikely to have resulted from intravenous underdosing. While we already allude to this fact in the Discussion, we have now highlighted that future dose-response studies with intravenous oxytocin would be important to better clarify the impact that increases of oxytocin in systemic circulation might have on the brain, as outlined below.

Line 391: *“Fourth, the amount of oxytocin we administered intravenously was not chosen to mimic exactly the plasmatic concentrations achieved after intranasal administration, which would require the infusion of a dose about 5 times lower (2IU)³⁷. Instead, we adopted a proof-of-concept approach, aiming to achieve consistently higher plasmatic concentrations of oxytocin during the full period of scanning, eliminating the hypothesis that negative findings could be ascribed to insufficient dosing. Future studies should compare the effects of different doses administered intravenously, including those mirroring the exact increases in plasmatic oxytocin produced by intranasal methods of administration.”*

9. It is not clear from the text when drug administration was relative to blood sampling. Please clarify the exact timeline of the sessions.

10. Figure 1 needs more clarity. The numbers underneath presumably represent when blood samples were taken (this could be made clearer), but it would also be helpful to see explicitly at what time after drug administration these were (on average). For example, sample 3 was taken, immediately after drug administration (or 6 mins depending on the session?), but when was sample 4 taken etc? Was sample 1 the baseline for every participant regardless of session? From the text one is able to infer that 6, 10 & 6 refer to the duration of drug administration, but this could also be made more explicit. VAS presumably stands for Visual Analogue Scale, but this is not referenced in the figure caption.

Thank you for the opportunity to clarify these points and our figure. Here, we respond to both questions 9 and 10, since they are related. We have now revised figure 8 (previously figure 1) and the respective caption as suggested by the reviewer to add clarity, according to the reviewer's suggestions. The times at which blood samples were obtained are expressed in minutes, with the end of the last treatment administration as the reference point (time 0). A blood sample was obtained at baseline (before introducing any treatment), and at the end of treatment administrations 2 and 3 (which could contain active oxytocin). We have now added labels in the figure accordingly. We report times for blood samples and resting-state fMRI scan from the end of our last treatment administration (2nd intranasal administration of either placebo or oxytocin).

Just to clarify, all participants received administration using the three methods in all sessions. The first administration was an intranasal placebo (either spray or nebuliser), the second was intravenous saline or oxytocin (10 IU) and the third intranasal placebo or oxytocin (40 IU) (either spray or nebuliser). This means that, for instance, in the intravenous oxytocin condition, participants received first an intranasal placebo (using either the spray or nebuliser), then intravenous oxytocin and then another intranasal administration of placebo (using either the spray or nebuliser). Hence, the time between treatment administration and blood sample/scan is consistent across treatment levels for all participants. Sample 1 was always collected immediately before any treatment administration. Sample 2 was taken after the intravenous administration of saline/oxytocin and immediately before the start of our second intranasal administration of placebo/oxytocin; therefore, around 16 minutes after the baseline sample. Sample 3 was collected immediately after the end of our second intranasal administration of placebo/oxytocin, before participants entered the MRI scanner; hence, around 22 minutes after the baseline sample. The next 4 samples were collected during the MRI scan, at fixed time-intervals (i.e. immediately before the first ASL scan, immediately after the second ASL scan, immediately before the start of the resting-state fMRI scan we

Department of Neuroimaging

report here and immediately before our 7th ASL scan). The approximate times from our last treatment administration each sample was collected have now been added to the figure. Immediately after the end of the MRI session, we collected the last blood sample (hence at around 104 minutes post last treatment offset).

“Fig. 8 – Study design, drug administration and blood sampling procedures (NB – Nebuliser; IV – Intravenous; PLC – Placebo; pK – Pharmacokinetics; RSF – Resting-state BOLD fMRI; VAS – Visual Analog Scale).

Upper panel – Drug allocation scheme: In each session, participants received treatment via all three administration routes, in one of two fixed sequences: either nebuliser/intravenous infusion/standard nasal spray, or standard nasal spray/intravenous infusion/nebuliser. In three out of four sessions only one route of administration contained the active drug; in the fourth session, all routes delivered placebo or saline. Unbeknown to the participants, the first treatment administration method in each session always contained placebo, while intranasal (spray or nebuliser) oxytocin was only delivered with the third treatment administration. The second administration was an infusion of either saline or oxytocin.

Lower panel - Study protocol: After drug administration, participants were guided to the MRI scanner, where eight pulsed continuous arterial spin labelling scans (results reported elsewhere¹⁶) and one resting-state BOLD-fMRI (RSF) were acquired, spanning 15-104 minutes after last drug administration offset. We assessed participants’ levels of alertness and excitement using visual analogue scales (VAS) at three different time-points during the scanning session to evaluate subjective drug effects across time (results reported elsewhere¹⁶). Intranasal administrations lasted for about 6 minutes, while the intravenous administration occurred at the highest rate of IU/min for 10 minutes. We collected plasma samples at baseline and at five time points post-dosing to measure changes in the concentration of oxytocin. The resting-state BOLD-fMRI scan used in this study was acquired between 57 – 65 minutes post last drug administration offset (the end of our second intranasal administration was considered time = 0). *Duration of treatment administration in minutes.”

11. I haven’t personally come across the idea of a word cloud to present the potential behavioural relevance of oxytocin’s effects on connectivity, but I liked the idea and found it very interesting.

Thanks very much.

12. It seems strange to anticipate different head movements based on drug condition, has this been found previously? If not, perhaps move the whole analysis to supplementary materials and briefly mention in the data pre-processing subsection.

**Department of
Neuroimaging**

Thanks for the suggestion. Indeed, effects of oxytocin on head movement have not been reported before. However, given that head movement can have a strong impact on resting-state BOLD-fMRI connectivity measures³⁸, we decided to be conservative and investigate treatment effects on head movement to discard this potential confounder. The lack of treatment effects on head movement would increase our certainty that our treatment effects on connectivity cannot be attributed to differences between treatment levels in head motion, if treatment had unpredictable effects on head movement. In any case, this analysis was secondary; therefore, we decided to follow the reviewer's suggestion and have now moved this section to Supplementary.

13. In the result section the authors are explicit about which blood sample was used to compare effects of administration techniques on oxytocin concentrations, but it should also be stated explicitly in the methods section as well. Given that existing literature suggests there are individual differences in baseline oxytocin concentrations, did the authors consider using the baseline sample in their analyses? It would also be interesting to know whether AUC oxytocin up to sample 6 differed across treatment groups (from Fig 6.a of the authors other 2020 paper this doesn't seem likely, but as this analysis was run on the whole session not up to sample 6, it is not possible to say for certain).

Thanks for these suggestions. We have now made it explicit in the methods that we focused on the concentrations of plasmatic oxytocin at baseline and immediately before the resting-state BOLD fMRI scan we used for the analyses reported in this manuscript (see below).

Line 495: *"Here, we focused on the concentrations of plasmatic oxytocin at the sample we acquired at baseline and immediately before the resting-state fMRI scan (~ 57 minutes after the end of the last treatment administration). The full time-courses of the changes in plasmatic oxytocin after each method/route of administration can be found elsewhere¹⁶."*

We agree that including the baseline in these analyses would be important. Hence, we repeated our analyses considering baseline and redid Fig. 7 to include the baseline concentrations (please see below). Regarding the second suggestion of the reviewer, we have previously reported the areas under the curve (AUC) for the full time-course in each treatment condition earlier this year¹⁶. In this manuscript, we further confirm that i) plasmatic oxytocin remains elevated for all methods/routes of administration, as compared to placebo; ii) the intranasal methods do not differ in the increases in plasmatic oxytocin they achieved at the specific time-point our resting-state BOLD fMRI scan was acquired (57 mins after the end of our last treatment administration). As the reviewer highlights in this comment, the visual inspection of the time-courses of the concentrations of oxytocin in plasma for each method/route and the lack of differences in the comparison of AUC for these full time-courses do not suggest any difference in the plasmatic pharmacokinetics of oxytocin between intranasal methods of administration. In pharmacokinetics analyses, AUCs are typically calculated from time 0 to infinity. This was the approach we took to calculate the AUCs we reported earlier. Hence, we believe including this partial AUC in the current manuscript is probably redundant and might even create confusion to the reader when two different AUCs are calculated using two different approaches in two different manuscripts. Therefore, here we would rather keep the focus on sample 6. Nevertheless, we are happy to follow any further advice the reviewer/editor might have on this respect.

Methods

Line 626: *"Finally, we wanted to understand whether the plasmatic oxytocin concentrations at the time the resting-state scan was acquired (57 minutes after the end of the last treatment administration) differed across the four treatment conditions. For this purpose, we conducted a linear mixed model (implemented in SPSS) on plasmatic oxytocin concentrations measured in a blood sample collected immediately before the beginning of the resting-state acquisition, using Time (baseline, oxytocin concentration at 57 min post-administration), Treatment and Time x Treatment as fixed factors and a random intercept for subject. We followed up the significant interaction with further post-hoc tests to examine differences between each pair of treatment conditions between*

**Department of
Neuroimaging**

baseline and post-administration and between treatment conditions within each time point, using Tukey's correction for multiple testing."

Results

Line 211: "We found a significant interaction Time x Treatment in plasma oxytocin concentration ($F(3, 46.686) = 25.550, P < 0.0001$). Post-hoc explorations revealed that, at baseline, the four treatment conditions did not differ in plasma oxytocin concentration (minimal $p_{Tukey} = 0.540$). Intranasal and intravenous administrations of oxytocin resulted in increased plasma oxytocin concentration at 57 min post-administration, compared to baseline (all $p_{Tukey} < 0.010$), but the concentration of plasma OT in the placebo condition did not change significantly between the two time points ($p = 0.460$). Plasmatic oxytocin concentrations at post-administration were higher in the intravenous condition than in all the other three treatment conditions (all $p_{Tukey} < 0.010$). Spray and nebuliser did not differ in plasma oxytocin concentrations at 57 min post-administration ($p_{Tukey} = 1.000$) (Fig. 7)."

Figures

"Fig. 7 – Baseline and post-administration (57 minutes from last treatment offset) concentrations of plasmatic oxytocin in each treatment condition. We conducted a linear mixed model on plasmatic oxytocin concentrations measured in a blood sample collected immediately before the beginning of the resting-state acquisition, using Time, Treatment and Time x Treatment as fixed factors and a random intercept for subject. We followed up the significant interaction with further post-hoc tests to examine differences between each pair of treatment conditions between baseline and post-administration and between treatment conditions within each time level, using Tukey's correction for multiple testing. In the upper panel, we present box and violin plots depicting the distribution of plasmatic concentrations of oxytocin for each treatment condition and time point; middle horizontal lines represent the mean; boxes indicate the 25th and 75th percentiles. In the lower panel, we present the descriptive statistics for each level of the two factors. *Intravenous condition compared to the other three ($p < 0.01$) at post-administration; # Spray and Nebuliser conditions compared to placebo (< 0.001) at post-administration."

Time	Treatment	Concentration (pg/ml)	Standard error of the mean
Baseline	Spray	3.701	0.757
	Nebuliser	2.689	0.589
	Intravenous	2.763	0.565
	Placebo	2.345	0.528
57 minutes post-administration	Spray	7.034	0.929
	Nebuliser	7.207	1.151
	Intravenous	16.35	1.802
	Placebo	1.577	0.234

**Department of
Neuroimaging**

Minor comments:

Pg. 3, line 69 – Should read “Although these types of studies are...”

The sentence has now been edited.

Pg. 10, line 336 – Should read “... these differences between the...”

Thanks. The typo has been corrected.

Pg. 12 – In the PARI SINUS nebuliser subsection, missing a % mark after 15.7

The typo has been corrected.

Pg. 15 - missing a % mark after 5

The typo has been corrected.

Pg. 16 – In the Mesoscale subsection, don't need to repeat the final sentence, as this is already stated in the overview.

Thanks for pointing that out. We have now removed the last sentence.

Supplementary materials, Table S1 – delete ‘ed’ from regions-of-interested.

The typo has now been corrected.

References

- 1 Andellini, M., Cannata, V., Gazzellini, S., Bernardi, B. & Napolitano, A. Test-retest reliability of graph metrics of resting state MRI functional brain networks: A review. *J Neurosci Methods* **253**, 183-192, doi:10.1016/j.jneumeth.2015.05.020 (2015).
- 2 Welton, T., Kent, D. A., Auer, D. P. & Dineen, R. A. Reproducibility of graph-theoretic brain network metrics: a systematic review. *Brain Connect* **5**, 193-202, doi:10.1089/brain.2014.0313 (2015).
- 3 Termenon, M., Jaillard, A., Delon-Martin, C. & Achard, S. Reliability of graph analysis of resting state fMRI using test-retest dataset from the Human Connectome Project. *Neuroimage* **142**, 172-187, doi:10.1016/j.neuroimage.2016.05.062 (2016).
- 4 Kolossa, A. & Kopp, B. Data quality over data quantity in computational cognitive neuroscience. *Neuroimage* **172**, 775-785, doi:10.1016/j.neuroimage.2018.01.005 (2018).
- 5 Dipasquale, O. *et al.* Comparing resting state fMRI de-noising approaches using multi- and single-echo acquisitions. *PLoS One* **12**, e0173289, doi:10.1371/journal.pone.0173289 (2017).
- 6 Shamay-Tsoory, S. G. & Abu-Akel, A. The Social Salience Hypothesis of Oxytocin. *Biol Psychiatry* **79**, 194-202, doi:10.1016/j.biopsych.2015.07.020 (2016).
- 7 Harari-Dahan, O. & Bernstein, A. A general approach-avoidance hypothesis of oxytocin: accounting for social and non-social effects of oxytocin. *Neurosci Biobehav Rev* **47**, 506-519, doi:10.1016/j.neubiorev.2014.10.007 (2014).

**Department of
 Neuroimaging**

- 8 Quattrocki, E. & Friston, K. Autism, oxytocin and interoception. *Neurosci Biobehav Rev* **47**, 410-430, doi:10.1016/j.neubiorev.2014.09.012 (2014).
- 9 Xin, F. *et al.* Oxytocin Modulates the Intrinsic Dynamics Between Attention-Related Large-Scale Networks. *Cereb Cortex*, doi:10.1093/cercor/bhy295 (2018).
- 10 Brodmann, K., Gruber, O. & Goya-Maldonado, R. Intranasal Oxytocin Selectively Modulates Large-Scale Brain Networks in Humans. *Brain Connect* **7**, 454-463, doi:10.1089/brain.2017.0528 (2017).
- 11 Yeo, B. T. *et al.* The organization of the human cerebral cortex estimated by intrinsic functional connectivity. *J Neurophysiol* **106**, 1125-1165, doi:10.1152/jn.00338.2011 (2011).
- 12 Jong, T. R. *et al.* Salivary oxytocin concentrations in response to running, sexual self-stimulation, breastfeeding and the TSST: The Regensburg Oxytocin Challenge (ROC) study. *Psychoneuroendocrinology* **62**, 381-388, doi:10.1016/j.psyneuen.2015.08.027 (2015).
- 13 Aulinas, A. *et al.* Endogenous Oxytocin Levels in Relation to Food Intake, Menstrual Phase, and Age in Females. *The Journal of clinical endocrinology and metabolism* **104**, 1348-1356, doi:10.1210/jc.2018-02036 (2019).
- 14 Perry, R. N. *et al.* The impacts of actual and perceived nicotine administration on insula functional connectivity with the anterior cingulate cortex and nucleus accumbens. *Journal of psychopharmacology (Oxford, England)* **33**, 1600-1609, doi:10.1177/0269881119872205 (2019).
- 15 Goriounova, N. A. & Mansvelder, H. D. Short- and long-term consequences of nicotine exposure during adolescence for prefrontal cortex neuronal network function. *Cold Spring Harb Perspect Med* **2**, a012120, doi:10.1101/cshperspect.a012120 (2012).
- 16 Martins, D. A. *et al.* Effects of route of administration on oxytocin-induced changes in regional cerebral blood flow in humans. *Nat Commun* **11**, 1160, doi:10.1038/s41467-020-14845-5 (2020).
- 17 Bullmore, E. & Sporns, O. Complex brain networks: graph theoretical analysis of structural and functional systems. *Nat Rev Neurosci* **10**, 186-198, doi:10.1038/nrn2575 (2009).
- 18 Salonia, A. *et al.* Menstrual cycle-related changes in plasma oxytocin are relevant to normal sexual function in healthy women. *Horm Behav* **47**, 164-169, doi:10.1016/j.yhbeh.2004.10.002 (2005).
- 19 Ruis, C. *et al.* Symptom Checklist 90-Revised in neurological outpatients. *J Clin Exp Neuropsychol* **36**, 170-177, doi:10.1080/13803395.2013.875519 (2014).
- 20 Sacco, R. *et al.* Psychometric properties and validity of Beck Depression Inventory II in multiple sclerosis. *Eur J Neurol* **23**, 744-750, doi:10.1111/ene.12932 (2016).
- 21 Francis, S. M., Kirkpatrick, M. G., de Wit, H. & Jacob, S. Urinary and plasma oxytocin changes in response to MDMA or intranasal oxytocin administration. *Psychoneuroendocrinology* **74**, 92-100, doi:10.1016/j.psyneuen.2016.08.011 (2016).
- 22 Kirkpatrick, M. G., Francis, S. M., Lee, R., de Wit, H. & Jacob, S. Plasma oxytocin concentrations following MDMA or intranasal oxytocin in humans. *Psychoneuroendocrinology* **46**, 23-31, doi:10.1016/j.psyneuen.2014.04.006 (2014).
- 23 Fafrowicz, M. *et al.* Beyond the Low Frequency Fluctuations: Morning and Evening Differences in Human Brain. *Front Hum Neurosci* **13**, 288, doi:10.3389/fnhum.2019.00288 (2019).
- 24 Kagerbauer, S. M. *et al.* Absence of a diurnal rhythm of oxytocin and arginine-vasopressin in human cerebrospinal fluid, blood and saliva. *Neuropeptides* **78**, 101977, doi:10.1016/j.npep.2019.101977 (2019).
- 25 Paloyelis, Y. *et al.* A Spatiotemporal Profile of In Vivo Cerebral Blood Flow Changes Following Intranasal Oxytocin in Humans. *Biol Psychiatry* **79**, 693-705, doi:10.1016/j.biopsych.2014.10.005 (2016).

**Department of
Neuroimaging**

- 26 Martins, D. *et al.* Intranasal oxytocin increases heart-rate variability in men at clinical high risk for psychosis: a proof-of-concept study. *Transl Psychiatry* **10**, 227, doi:10.1038/s41398-020-00890-7 (2020).
- 27 Schmidt, A. *et al.* Acute oxytocin effects in inferring others' beliefs and social emotions in people at clinical high risk for psychosis. *Transl Psychiatry* **10**, 203, doi:10.1038/s41398-020-00885-4 (2020).
- 28 Martins, D. *et al.* Investigating resting brain perfusion abnormalities and disease target-engagement by intranasal oxytocin in women with bulimia nervosa and binge-eating disorder and healthy controls. *Transl Psychiatry* **10**, 180, doi:10.1038/s41398-020-00871-w (2020).
- 29 Leslie, M., Leppanen, J., Paloyelis, Y. & Treasure, J. A pilot study investigating the influence of oxytocin on attentional bias to food images in women with bulimia nervosa or binge eating disorder. *J Neuroendocrinol* **32**, e12843, doi:10.1111/jne.12843 (2020).
- 30 Leslie, M., Leppanen, J., Paloyelis, Y., Nazar, B. P. & Treasure, J. The influence of oxytocin on risk-taking in the balloon analogue risk task among women with bulimia nervosa and binge eating disorder. *J Neuroendocrinol* **31**, e12771, doi:10.1111/jne.12771 (2019).
- 31 Davies, C. *et al.* Neurochemical effects of oxytocin in people at clinical high risk for psychosis. *European neuropsychopharmacology : the journal of the European College of Neuropsychopharmacology* **29**, 601-615, doi:10.1016/j.euroneuro.2019.03.008 (2019).
- 32 Davies, C. *et al.* Oxytocin modulates hippocampal perfusion in people at clinical high risk for psychosis. *Neuropsychopharmacology* **44**, 1300-1309, doi:10.1038/s41386-018-0311-6 (2019).
- 33 Leslie, M., Leppanen, J., Paloyelis, Y. & Treasure, J. The influence of oxytocin on eating behaviours and stress in women with bulimia nervosa and binge eating disorder. *Molecular and cellular endocrinology* **497**, 110354, doi:10.1016/j.mce.2018.12.014 (2019).
- 34 Paloyelis, Y. *et al.* The Analgesic Effect of Oxytocin in Humans: A Double-Blind, Placebo-Controlled Cross-Over Study Using Laser-Evoked Potentials. *J Neuroendocrinol* **28**, doi:10.1111/jne.12347 (2016).
- 35 Cai, Q., Feng, L. & Yap, K. Z. Systematic review and meta-analysis of reported adverse events of long-term intranasal oxytocin treatment for autism spectrum disorder. *Psychiatry Clin Neurosci* **72**, 140-151, doi:10.1111/pcn.12627 (2018).
- 36 Guastella, A. J. *et al.* Recommendations for the standardisation of oxytocin nasal administration and guidelines for its reporting in human research. *Psychoneuroendocrinology* **38**, 612-625, doi:10.1016/j.psyneuen.2012.11.019 (2013).
- 37 Quintana, D. S. *et al.* Low dose intranasal oxytocin delivered with Breath Powered device dampens amygdala response to emotional stimuli: A peripheral effect-controlled within-subjects randomized dose-response fMRI trial. *Psychoneuroendocrinology* **69**, 180-188, doi:10.1016/j.psyneuen.2016.04.010 (2016).
- 38 Goto, M. *et al.* Head Motion and Correction Methods in Resting-state Functional MRI. *Magn Reson Med Sci* **15**, 178-186, doi:10.2463/mrms.rev.2015-0060 (2016).

REVIEWERS' COMMENTS:

Reviewer #1 (Remarks to the Author):

The authors sufficiently addressed my concerns

Reviewer #2 (Remarks to the Author):

All comments have been addressed.